# The transcription factor Hey and nuclear lamins specify and maintain cell identity

**Naama Flint Brodsly[1†], Eliya Bitman-Lotan[1†], Olga Boico[1], Adi Shafat[1], Maria Monastirioti[2], Manfred Gessler[3], Christos Delidakis[2], Hector Rincon-Arano[4], Amir Orian[1]\***

[1]Rappaport Research Institute and Faculty of Medicine, Technion-Israel Institute of Technology, Haifa, Israel; [2]Institute of Molecular Biology and Biotechnology (IMBB), Foundation for Research and Technology - Hellas (FORTH), Heraklion, Greece; [3]Biocenter of Developmental Biochemistry, University of Würzburg, Würzburg, Germany; [4]Division of Basic Sciences, Fred Hutchinson Cancer Research Center, Seattle, United States

**Abstract** The inability of differentiated cells to maintain their identity is a hallmark of age-related diseases. We found that the transcription factor Hey supervises the identity of differentiated enterocytes (ECs) in the adult *Drosophila* midgut. Lineage tracing established that Hey-deficient ECs are unable to maintain their unique nuclear organization and identity. To supervise cell identity, Hey determines the expression of nuclear lamins, switching from a stem-cell lamin configuration to a differentiated lamin configuration. Moreover, continued Hey expression is required to conserve large-scale nuclear organization. During aging, Hey levels decline, and EC identity and gut homeostasis are impaired, including pathological reprograming and compromised gut integrity. These phenotypes are highly similar to those observed upon acute targeting of Hey or perturbation of lamin expression in ECs in young adults. Indeed, aging phenotypes were suppressed by continued expression of Hey in ECs, suggesting that a Hey-lamin network safeguards nuclear organization and differentiated cell identity.
DOI: https://doi.org/10.7554/eLife.44745.001

**\*For correspondence:**
mdoryan@tx.technion.ac.il

[†]These authors contributed equally to this work

**Competing interests:** The authors declare that no competing interests exist.

## Introduction

Experiments such as nuclear transfer and reprogramming of differentiated fibroblasts into pluripotent cells (iPS) have changed the classical view of a rigid 'terminally-differentiated' cell state to a more plastic one (*Gurdon, 1962*; *Takahashi and Yamanaka, 2006*; *Morris, 2016*), suggesting that once established, differentiated cells must actively maintain their identities (*Blau and Baltimore, 1991*; *Natoli, 2010*; *Holmberg and Perlmann, 2012*; *Bitman-Lotan and Orian, 2018*). Indeed, failure to maintain a differentiated identity is associated withn altered physiological properties of post-mitotic cells and tissues, resulting in disease such as diabetes, neurodegeneration, and cancer (*Deneris and Hobert, 2014*; *Ocampo et al., 2016*; *Schwitalla et al., 2013*). Moreover, recently it was shown that loss of identity is a hallmark of the aging *Drosophila* midgut (*Li et al., 2016*).

Differentiated cells maintain their identity by multiple mechanisms, including tissue-specific transcription factors (TFs) and the control of high-order chromatin structure (e.g., *Cobaleda et al., 2007*; *Natoli, 2010*; *Holmberg and Perlmann, 2012*; *Lin and Murre, 2013*). Additionally, nuclear lamins are essential in establishing a nuclear organization that is unique to the differentiated state (*Kohwi et al., 2013*; *Gruenbaum and Foisner, 2015*). These mechanisms likely serve as a barrier against pathological reprograming and are highly relevant to human disease and regenerative medicine. While differentiated cells exhibit distinct chromatin and nuclear organization, the mechanisms

by which identity supervisors establish, maintain, and orchestrate these multi-levels of identity regulation are less clear.

One tissue used to study how differentiated cells maintain their identity in the context of a highly regenerating tissue in vivo is the *Drosophila* adult gut epithelium (*Figure 1A*; *Lemaitre and Miguel-Aliaga, 2013*; *Guo et al., 2016*). Intestinal stem cells (ISCs) proliferate within the adult gut epithelia of both flies and vertebrates to either self-renew or differentiate into mature differentiated gut cells. Differentiated gut cells are characterized by functional diversity and a short lifespan (*Jiang and Edgar, 2012*; *Neves et al., 2015*). One of the most studied *Drosophila* gut regions is the midgut, which is further divided into sub-regions. While each sub-region has specific characteristics, key regulatory principles are common along the entire midgut (*Marianes and Spradling, 2013*; *Dutta et al., 2015*). In the midgut epithelium, intestinal stem cells (ISCs) either self-renew or mature into enteroblasts (EBs, *Figure 1A*; *Micchelli and Perrimon, 2006*; *Ohlstein and Spradling, 2006*), which in turn differentiate into large absorptive polyploid enterocytes (ECs). A smaller population of ISCs differentiate into secretory enteroendocrine cells (pre-EE and subsequently EEs; *Beehler-Evans and Micchelli, 2015*; *Guo et al., 2016*; *Sallé et al., 2017*). As in vertebrates, *Drosophila* midgut homeostasis requires extensive signaling between different epithelial gut cells as well as crosstalk with cells in neighboring tissues. Well-conserved signaling pathways, such as Notch, Wnt, EGF, Jak-Stat, and JNK, govern stem cell differentiation and are activated by diverse physiological changes, such as regeneration upon injury or response to infection (*Biteau et al., 2008*; *Jiang et al., 2011*; *Ferrandon, 2013*). During aging, gut homeostasis is impaired, resulting in aberrant signaling, loss of EC physiology, and mis-differentiation of progenitor cells (*Biteau et al., 2008*; *Rera et al., 2012*; *Buchon et al., 2013*; *Li et al., 2016*; *Miguel-Aliaga et al., 2018*).

A central pathway that regulates ISC differentiation and gut homeostasis is the Notch pathway, which plays multiple roles in the midgut (*Bray, 2016*). High levels of Notch activity are required for ISC differentiation and the acquisition of an EC fate. Part of the Notch activity is mediated by the evolutionarily conserved HES (hairy/enhancer of split) and HES-related bHLH-transcription factors. In progenitor cells, HES limits ISC self-renewal and promotes differentiation by inhibiting the expression of the Notch ligand, Delta (*Bardin et al., 2010*; *Perdigoto et al., 2011*). Here we report that *Drosophila* Hey, a HES-related transcription factor, is a critical supervisor of ECs' differentiated identity (*Monastirioti et al., 2010*; *Guiu et al., 2013*; *Housden et al., 2013*). In vertebrates, Hey proteins (Hey1, Hey2, HeyL) regulate cell fate decisions during cardiogenesis, angiogenesis, and neurogenesis, as well as within the immune system (*Heisig et al., 2012*; *Weber et al., 2014*). The *Drosophila* genome encodes a single Hey protein (CG11194) that is required for embryonic development and larval neurogenesis (*Lu et al., 2008*; *Monastirioti et al., 2010*; *Zacharioudaki et al., 2012*). While a wealth of data has accumulated on Hey protein function during development, little is known regarding its function in differentiated cells within adult tissues. We found that Hey supervises the identity of fully differentiated ECs in the adult midgut by continued regulation of nuclear lamins expression. In concert with Lamin C (LamC, *Drosophila* A-type lamin), Hey maintains a nuclear architecture unique to the differentiated ECs. Moreover, misexpression of nuclear lamins in non-relevant cells overrides the endogenous cell identity programs. Remarkably, the level of Hey in ECs decline with aged and forced expression of Hey in aged ECs restores ECs identity, suppressing aging phenotypes. Thus, the continued supervision of chromatin and nuclear organization by Hey is critical for maintaining cell identity, tissue homeostasis, and organismal survival.

## Results

### Hey is required to maintain the differentiated state of ECs

Our initial step in assessing the role(s) of Hey in adult intestinal epithelia was to determine the protein expression of endogenous Hey in the gut epithelia cells using an anti-Hey antibody together with cell-specific GFP reporters and cell-specific markers (*Figure 1B–D*; *Figure 1—figure supplement 1K–P*; *Monastirioti et al., 2010*). We found that Hey protein is expressed in ISCs, EBs, EEs and is abundant in fully differentiated ECs, suggesting a functional role for Hey in ECs.

To address the function of Hey proteins in fully differentiated gut cells, we conditionally and temporally knocked down Hey in either ECs or EEs in 2–4 days old adult *Drosophila* females using several independent UAS-Hey-RNAi lines, under the control of cell-specific Gal4/Gal80$^{ts}$ coupled

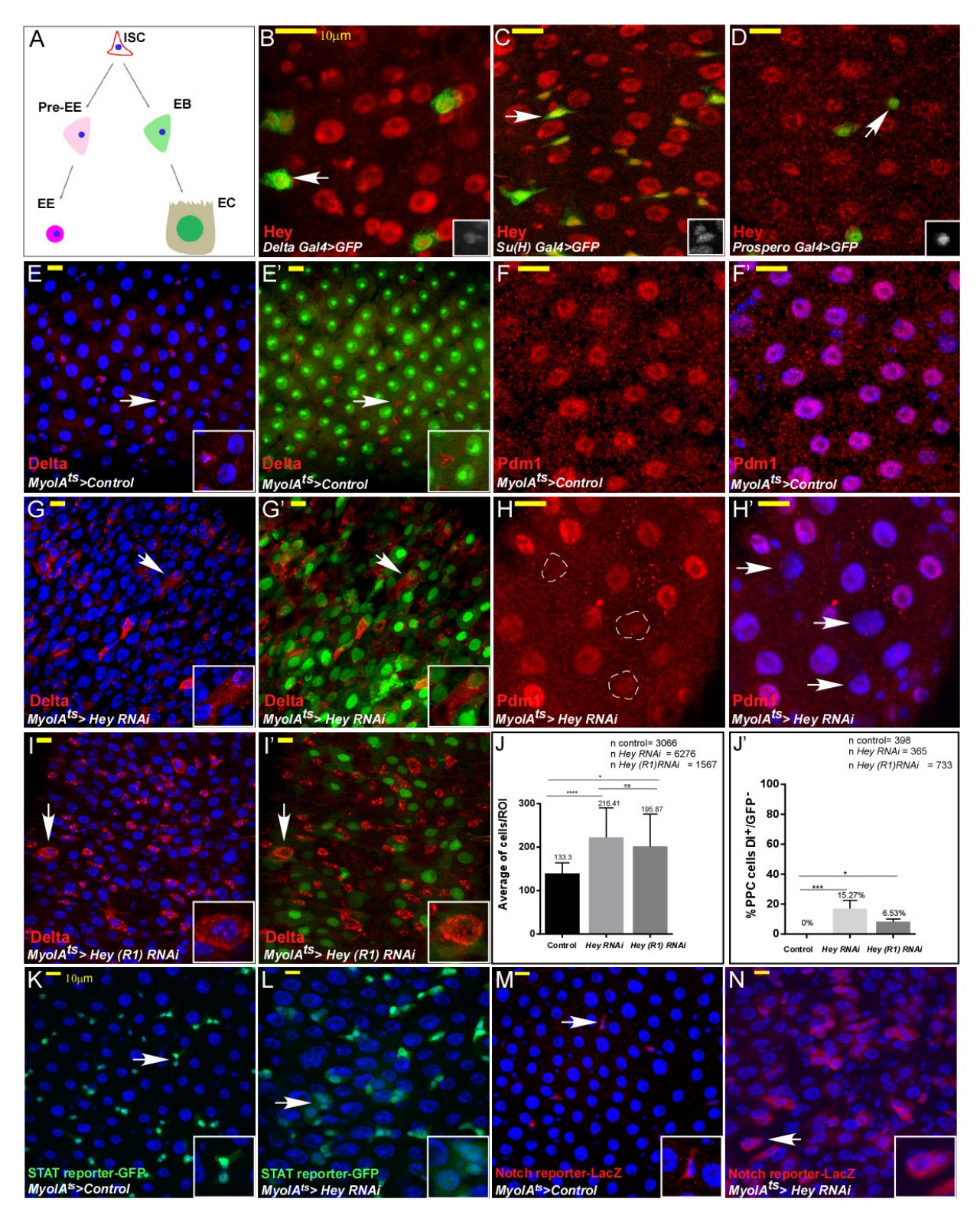

**Figure 1.** *Drosophila* Hey is required for maintaining EC identity. (**A**) Schematic diagram of major *Drosophila* midgut cell types. ISC, intestinal stem cell; EB, enteroblast; EC, enterocyte; Pre-EE, enteroendocrine progenitor, EE enteroendocrine cell. (**B–I'**) Confocal images of adult *Drosophila* midgut intestinal epithelium expressing the indicated transgenes, scale bar is 10 μm. Arrows indicate cells shown in insets. (**B–D**) Endogenous Hey protein was identified using an α-Hey antibody (red). The expression of UAS-GFP under the control of *Delta*-Gal4 (**B**), *Su(H)*-Gal4 (**C**), and *prospero*-Gal4 (**D**) mark

*Figure 1 continued on next page*

*Figure 1 continued*

ISCs, EBs, and EEs, respectively. (**E–I′**) Confocal images of control cells (**E–F′**) or midguts in which Hey was targeted for 48 hr in ECs using the indicated RNAi transgenic lines (**G–I′**). α-Delta (red E-E′ G-G′ I-I′), *MyoIA*ts80 >GFP (green) and anti-PDM1(red, (**F, F′, H, H′**) mark differentiated ECs; DAPI marks DNA (blue). White dashed circles in (**H**) are Pdm1-negative polyploid cells indicated in H′ by white arrows. (**J, J′**) Quantification of total cell number per region of interest (**J**), and polyploid cells that express Delta and are GFP negative in experimental setting similar to G, G′ and I, I′;(**J′**) ****p<0.0001, ***p<0.001, *p<0.05. (**K–N**) Hey depletion for 48 hr in ECs results in activation of stress and regeneration pathways; ectopic activation of JAK-STAT reporter (**K, L**) (*10XSTAT*::GFP reporter, GFP) and Notch pathway reporter in polyploid cells (**M, N**) (*ghd3*::LacZ reporter; RFP). White arrows point to cells shown in the inset.

DOI: https://doi.org/10.7554/eLife.44745.002

The following source data and figure supplements are available for figure 1:

**Source data 1.** Quantification of average number off cells in control and guts where Hey is targeted in EC data related to *Figure 1*.
DOI: https://doi.org/10.7554/eLife.44745.007
**Figure supplement 1.** Analysis of Hey targeted ECs and EEs.
DOI: https://doi.org/10.7554/eLife.44745.003
**Figure supplement 1—source data 1.** Quantification data related to *Figure 1—figure supplement 1*.
DOI: https://doi.org/10.7554/eLife.44745.004
**Figure supplement 2.** Impact of RNAi-dependent loss of Hey in males and on the entire gut in females.
DOI: https://doi.org/10.7554/eLife.44745.005
**Figure supplement 3.** The effects on EC identity upon RNAi-dependent targeting of Notch, Su(H), HES transcription factors, and related cofactors.
DOI: https://doi.org/10.7554/eLife.44745.006

systems (see Materials and methods and figures for specific lines used; *Salmeron et al., 1990*; *Brand and Perrimon, 1993*). We used tub-Gal80ts; *MyoIA*-Gal4 (termed MyoIA^ts) to target Hey in ECs;, and the *Prospero*-Gal4; tub-Gal80^ts to target Hey in EEs (*Buchon et al., 2013*). RNAi knockdown of Hey in ECs and EEs had no detectable effect on either EE number or differentiation state, as indicated by the expression of the EE-related transcription factor Prospero (*Figure 1*-figure - However, upon conditional inactivation of Hey in adult ECs, expression of an EC marker *MyoIA*::GFP and the EC founder homeobox transcription factors Pdm1 and Odd-skipped (Odd) were reduced throughout the entire midgut (compare *Figure 1E–F′* with 1 G-H′, *Figure 1—figure supplement 1A–B′*, and quantified in *Figure 1—figure supplement 1E,F*; *Korzelius et al., 2014*; *Dutta et al., 2015*).

A second prominent phenotype was hyperplasia of intestinal epithelial cells, doubling the number of cells upon loss of Hey in EC (*Figure 1I–J′*). A third phenotype was the ectopic expression of the Notch ligand Delta, which is normally expressed only in ISCs and immature EE cells (*Figure 1J,J′*). This ectopic Delta expression was observed on the surface of large polyploid cells resembling ECs, in part not expressing *MyoIA*::GFP (*Figure 1G–I′* and quantified 1J, J′; *Guo et al., 2016*). These cells, however, did not express the EE marker Prospero, suggesting that they did not transdifferentiate to EEs (*Figure 1—figure supplement 1G,H*). The loss of MyoIA::GFP expression, ectopic Delta expression and overall gut morphology phenotypes were also observed in males, along the entire midgut (*Figure 1—figure supplement 2A–D′*), and were suppressed by over-expression of UAS-Hey, but not by a control transgene, (*Figure 1—figure supplement 1C,C′*). Moreover, RNAi knockdown of Notch, Suppressor of Hairless (Su(H)), other HES TFs, and HES-related cofactors did not phenocopy Hey loss in ECs (*Figure 1—figure supplement 3*) RNA-mediated knockdown of Hey in ECs not only impaired EC identity but also affected the entire epithelial tissue and gut homeostasis. Using pathway-specific reporter transgenes, we observed that loss of Hey in ECs resulted in hyper-activation of stress and regeneration-related pathways such as JAK/STAT and Notch including in polyploid cells (PPCs) in which these pathways are normally silent (*Figure 1K–N* and see also Figure 6). Thus, we conclude that maintaining EC identity and tissue homeostasis requires continuous Hey expression in ECs.

## Hey loss in ECs impairs EC identity and progenitor differentiation

We hypothesized that the above-mentioned phenotypes may originate either from inability of ECs to maintain differentiated identity, and/or due to activation of stress response of ISCs resulting in enlarged progenitors, that fail to fully differentiate into ECs in the absence of Hey (*Shaw et al., 2010*). Therefore, to determine the fate of individually targeted ECs and to assess the cellular

composition of the gut upon Hey loss in ECs, we used 'G-TRACE', a method for lineage tracing of non-dividing cells (*Figure 2A*, see Materials and methods; *Evans et al., 2009*). In brief, a UAS-RFP marker is expressed via the EC-specific promoter/driver *MyoAI*-Gal4, which is active only in fully differentiated ECs (red). The same *MyoIA*-Gal4 also drives the expression of a UAS-flipase that induces a recombination event that activates permanent GFP expression, which serves as a 'history marker'. This GFP 'history marker' is expressed in fully differentiated ECs and their progeny regardless of the cell's current differentiation state. Indeed, <u>all</u> ECs in control guts were both RFP and GFP positive (RFP$^+$, GFP$^+$) and appeared orange (*Figure 2B*, and quantitated in 2F). In contrast, twenty-four and forty-eight hours after Hey targeting in ECs, we observed diverse fluorescence populations of polyploid cells (PPCs; *Figure 2C–F*, and dynamic analysis is shown in *Figure 2—figure supplement 1A–A''''*): After forty-eight hours 48% of PPCs expressed both RFP$^{(+)}$, and GFP$^{(+)}$ (similar to control EC, *Figure 2C1*). 37% expressed only GFP, but not RFP (PPC$^{GFP(+), RFP(-)}$ termed PPC**; *Figure 2C'2*). We also observed 11% PPCs that were negative for both RFP and GFP (PPC $^{GFP(-), RFP(-)}$) and were characterized by large, polyploid nuclei (termed PPC*; *Figure 2C'3*). Finally, we detected few (4%) PPCs that were RFP$^{(+)}$, GFP$^{(-)}$; *Figure 2C'4*), likely reflecting ECs in which the recombination event did not take place. By the nature of the G-TRACE tracing lineage system we suggest that PPC are highly similar to control ECs, PPC** are former ECs that no-longer express the MyoIA >RFP and are likely former EC that no longer maintain EC identity. Likely PPC* are rapidly developing mis-differentiated progenitors/young PPCs that did not activate the marking system. Using G-TRACE, we further characterized the properties of the different PPCs. We found that PPC** cells as well as PPC* ectopically expressed the ISC marker Delta (*Figure 2E,E'*). Moreover, PPC** did not express the enterocyte founder transcription factor Pdm1, nor they express EC-specific genes like Odd-skipped and Snakeskin, a septate junction-related protein (*Figure 2G–I'*, *Supplementary file 1*; *Supplementary file 2*; *Korzelius et al., 2014*; *Dutta et al., 2015*). In addition, the overall ploidy of PPCs cells was only minimally affected (*Figure 2—figure supplement 1B*). Based on the nature of the G-TRACE system, these data suggest that Hey depletion in ECs resulted in EC that can no longer maintain EC identity (PPC**).

## Transcriptional regulation of EC identity by Hey

The inability to supervise EC identity is likely due to a failure in maintaining a Hey-related transcriptional program. We therefore used gene expression arrays to determine the transcription signature of whole guts upon targeting Hey in ECs using MyoIA-Gal4 and UAS-Hey RNAi. We also determined Hey-regulated genes in progenitors by comparing the gene signature of affinity-purified progenitors expressing either UAS-GFP (control) or UAS-Hey RNAi. Activation of the UAS-transgenes in progenitors was driven by *hey*-Gal4, which is expressed predominantly in EBs, minimally in ISC, and not at all in ECs and EEs. (*Figure 3—figure supplement 1A–F'*, and see below and Materials and methods).

Upon Hey knockdown in ECs, we identified 370 differentially expressed genes (DEGs) whose expression in the gut is Hey dependent (termed 'Hey-regulated genes'; *Supplementary file 1*, for PCA plots see *Figure 3—figure supplement 4D,E*), and see Materials and methods). Note that ~50% (113/228) of genes exhibiting reduced expression upon Hey targeting in ECs are genes that are repressed by the ectopic expression of the progenitor transcription factor Escargot (Esg) in ECs suggesting that they are EC-related (*Figure 3—figure supplement 1G*, *Supplementary file 5*; *Karolchik et al., 2014*). Overall, the transcriptional signature of guts in which Hey knockdown was induced in ECs largely resembled the transcriptional signature of control purified progenitors (see below *Figure 3A,A'*, *Supplementary file 1*). Gene ontology analyses revealed that Hey maintains a differentiated EC gene signature. It is required for the expression of numerous genes involved in EC physiology and metabolism, and numerous putative digestive enzymes such as genes involved in lipid transporters, lipase activity, amino acid metabolism, and peptidases (*Figure 3B*, *Figure 3—figure supplement 1Q*, *Supplementary file 2*; *Supplementary file 6*, https://flygut.epfl.ch/expressions).

Concomitantly, loss of Hey in ECs resulted in ectopic expression of pathways associated with nuclear and DNA-related processes (such as TFs, DNA replication/repair, ncRNAs, and nuclear organization (*Figure 3C*, *Figure 3—figure supplement 3H*, *Supplementary file 2*). The origin of these upregulated DEGs likely reflects PPC** and PPC* expression signatures, as well as an increase in

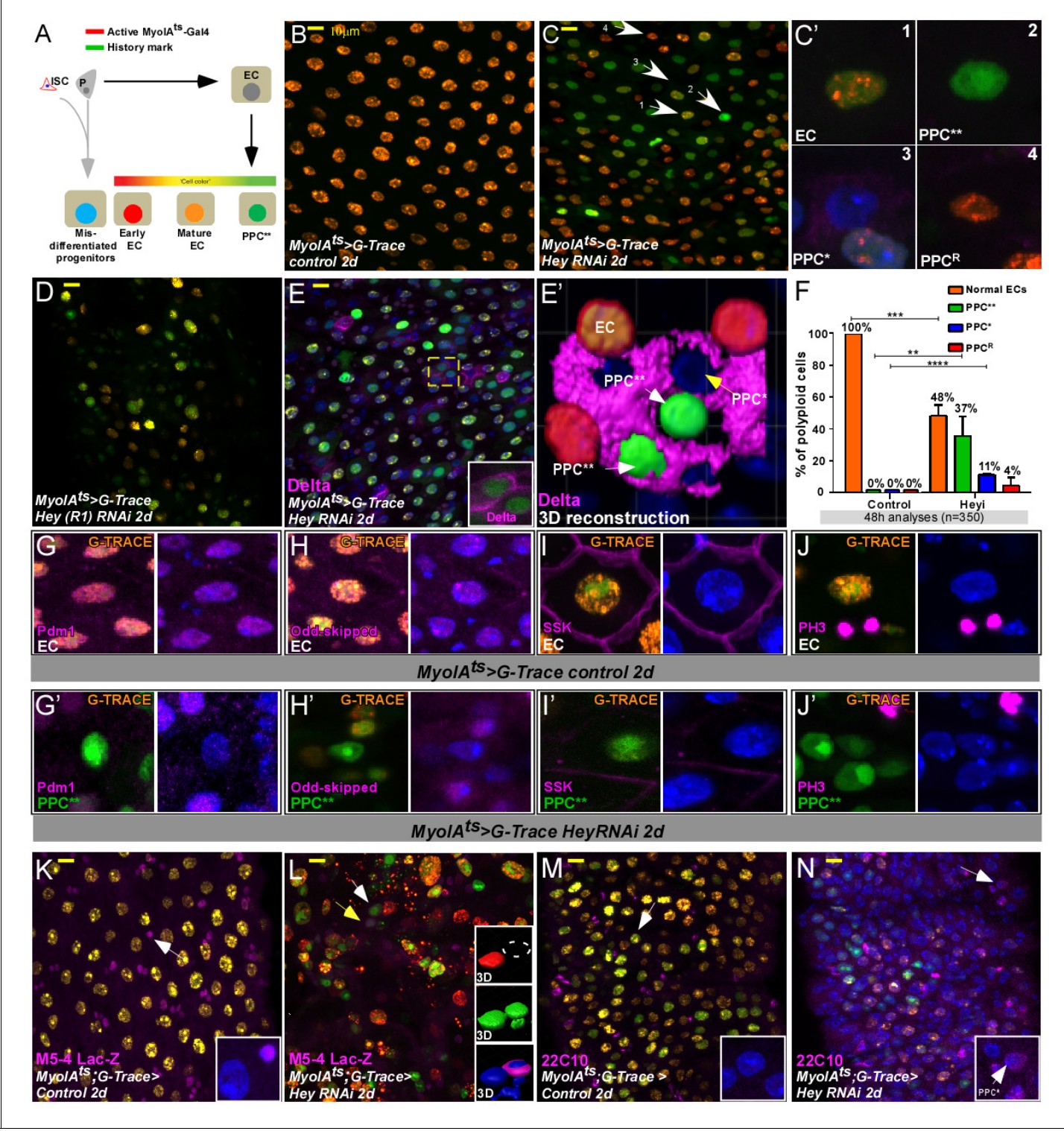

**Figure 2.** Gut analyses using G-TRACE lineage tracing of ECs. (A) Illustration of G-TRACE lineage tracing (adopted from *Evans et al., 2009*). (B–E')
Confocal images of G-TRACE analyses of control (B) and Hey-targeted ECs (C–E'). (B) All ECs in control guts co-express *MyoIA*-Gal4 >UAS RFP, and
the history marker Ub::GFP, and are denoted in orange [GFP $^{(+)}$ RFP $^{(+)}$]. (C–D) A heterogeneous population of polyploid cells is observed in midguts in
which Hey is targeted in ECs for 48 hr using the indicated Hey-RNAi transgenic lines. Numbered white arrows in C point to cells shown in C':
Differentiated ECs (C'1, GFP$^{(+)}$, RFP$^{(+)}$); PPC**, C'2, GFP$^{(+)}$RFP$^{(-)}$); PPC*, C'3, GFP$^{(-)}$ RFP$^{(-)}$); Enterocytes in which GFP-activating recombination has not
taken place (PPC$^R$ C'4, GFP$^{(-)}$RFP$^{(+)}$). DAPI marks DNA (blue). (E, E') PPC* and PPC** ectopically express Delta. (E') IMARIS-assisted 3D reconstruction of
the area indicated in E by a yellow square. A PPC** is indicated by a white arrow in E' and is shown in the inset of E. (F) G-TRACE-based quantification
*Figure 2 continued on next page*

*Figure 2 continued*

of polyploid cells after 48 hr; four independent biological repeats (n = 350, results shown are mean ± SD). (**G–J′**) G-TRACE-analyses of ECs (**G, H, I, J**) and PPC** (**G′ H′ I′ J′**). The indicated protein is shown in pink; (**G, G′**), Pdm1; (**H, H′**), Odd-skipped; (**I, I′**), SSK; (**J J′**) p-Histone-H3. DAPI marks DNA (blue). (**K–N**) Expression of the *escargot* progenitor enhancer reporter M5-4-LacZ (**K, L**) and neuronal marker 22C10 (**M, N**) in control or Hey-targeted ECs analyzed by G-TRACE, and see quantification ***Figure 4—figure supplement 3E***.

DOI: https://doi.org/10.7554/eLife.44745.008

The following source data and figure supplements are available for figure 2:

**Source data 1.** Quantification data related to ***Figure 2F***- % of PPC (EC/PPC**/PPC*/PPC^R).

DOI: https://doi.org/10.7554/eLife.44745.011

**Figure supplement 1.** Dynamic G-TRACE and ploidy analysis: (**A–A″″**) Analysis of endogenous Hey levels in control or Hey-targeted ECs using G-TRACE.

DOI: https://doi.org/10.7554/eLife.44745.009

**Figure supplement 1—source data 1.** Quantification data related to ***Figure 2—figure supplement 1***.

DOI: https://doi.org/10.7554/eLife.44745.010

rapidly dividing progenitors, as well as other cell populations that were newly present in the targeted gut (***Figure 3D–G′***).

We validated these results for a subset of genes at the protein level. For example, Delta, *maelstrom* (***Figure 3—figure supplement 1I–J***), and *γ-tubulin* (***Figure 3—figure supplement 1K–L***), which are normally repressed by Hey, were ectopically upregulated in polyploid cells (***Figure 1G, G′, I and I′***, ***Figure 3—figure supplement 1I–P***, ***Supplementary file 1***). Bona fide EC-specific genes *pdm1, odd-skipped, snakeskin, MESH, fasciclin,* which require Hey for their expression, exhibited reduced expression (see examples in ***Figures 1F,H*** and ***2G,G′*** (Pdm1); 2H, H′, ***Figure 1—figure supplement 1D,D′***, ***Figure 3—figure supplement 1M,N*** (Odd); 2I, I′ (SSK); 6E, F (MESH); Figure 6—figure supplement 1 (Fas, SSK, Dlg); ***Supplementary file 1***). These gene signatures are consistent with other studies that depict cell-specific gene profiles of midgut cells and regulators (***Supplementary file 6***; ***Korzelius et al., 2014***); https://flygut.epfl.ch/expressions). Moreover, loss of Hey resulted in the ectopic expression of progenitors and non-gut-related programs such as neurogenesis (12 genes, ***Supplementary file 2***), which is exemplified by the ectopic expression of the neuronal marker 22C10 in PPCs* (PPCs ^{GFP(-) RFP(-)}) cells (***Figure 2M,N***; ***Figure 3—figure supplement 1O,P***). Targeting Hey in ECs also effected the cellular composition of the gut and resulted in the emergence of new cell populations (***Figure 3D–G′***). For example, RecQ4, a Hey-regulated target gene (***Supplementary file 1***), is a DNA helicase that is co-expressed with LamC in fully differentiated ECs but not in ISCs (***Figure 3D and F***). Following Hey knockdown in ECs, we observed small cells expressing only RecQ4, without LamC but with Delta (RecQ4^{(+)}, Delta^{(+)}, LamC^{(-)}). We also observed that 17% of PPCs are RecQ4^{(+)}, Delta^{(+)} (***Figure 3E and G,G′***). These population of cells were not observed in control guts.

In parallel, we identified a group of Hey-regulated genes in progenitors (predominantly EBs) using an enriched affinity purification protocol. To identify Hey-regulated genes in progenitors, we expressed the CD8-GFP molecule on the surface of progenitors using the Hey-Gal4 (that is expressed predominantly EBs), along with control (UAS-GFP) or UAS-Hey RNAi (see detailed in Materials and methods). This surface labeling enabled us to affinity purify progenitors via CD8 magnetic beads and to compare the gene expression signature of control or Hey RNAi purified progenitors. We observed that three hundred and forty-five genes exhibited Hey-dependent regulation and were involved in progenitor maturation and EB identity (***Supplementary file 1***; ***Supplementary file 2***). We found that the expression signature of Hey in ECs is unique and overlaps only minimally with Hey-regulated genes in progenitors with only 10% (40) shared genes, suggesting cell-specific functions of Hey (***Figure 3H***). The transcriptional analysis of Hey-regulated genes in progenitors is consistent with our genetic experiments: upon cell-specific RNAi reduction of Hey in either ISC, EB, or both, the number of ECs declined and many progenitor cells were unable to maintain proper identity (***Figure 3—figure supplement 2***). In addition, *hey*-deficient clones generated by MARCM did not survive and were likely rapidly outcompeted (***Figure 3—figure supplement 3A–D***). In accordance, forced expression of Hey in progenitor cells resulted in the rapid differentiation of progenitors to PPCs, and ectopic endoreplication of *esg*::GFP^+ progenitors together with reduced Delta expression (but not Pdm1 expression) (***Figure 3—figure supplement 3E,F*** and not shown).

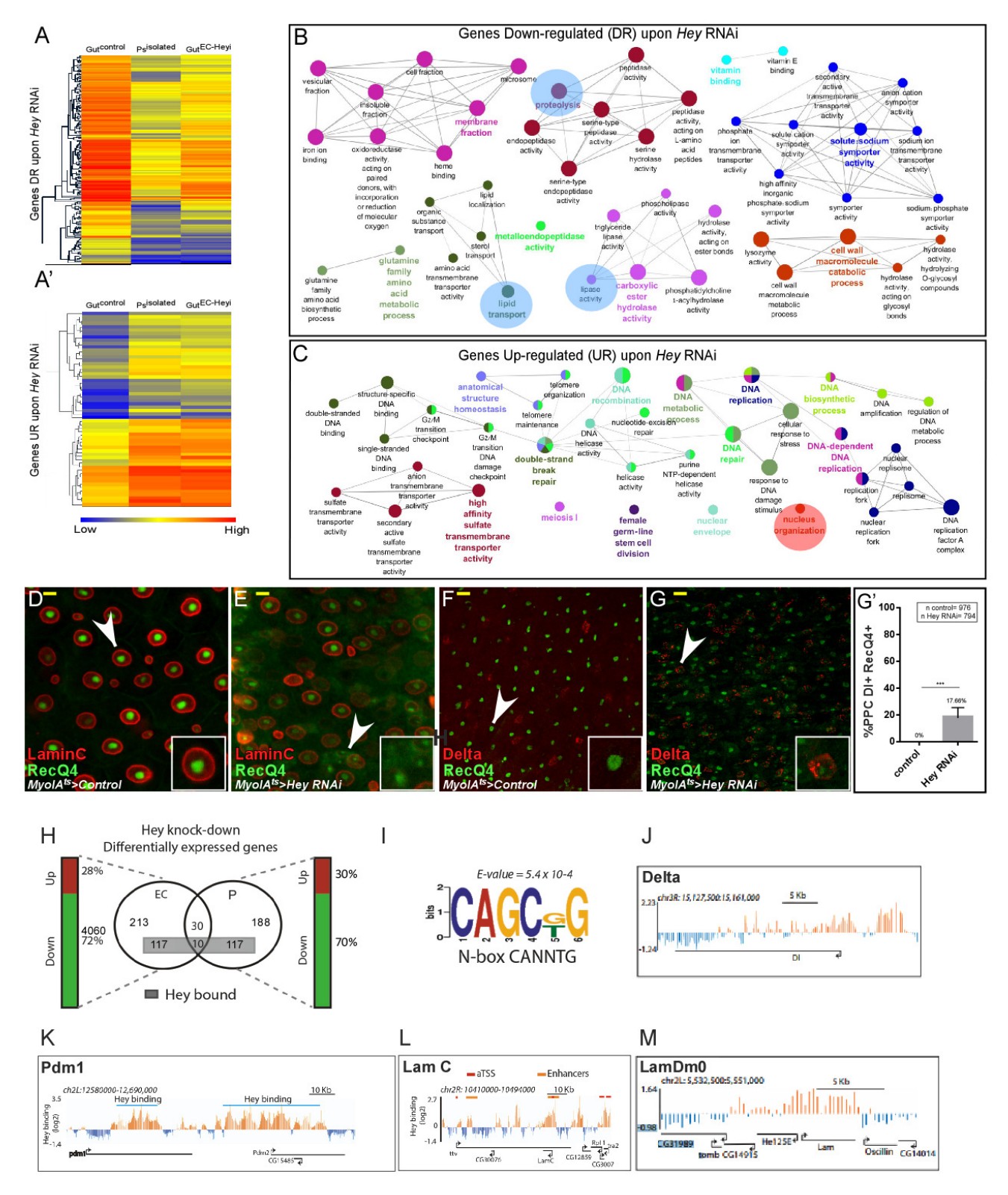

**Figure 3.** Transcriptional analysis of targeting Hey in ECs. (**A–A'**) Heat map depicting changes in gene expression in control guts, enriched purified progenitors (Ps^isolated), and guts in which Hey is targeted in ECs for 48 hr (Gut EC^Heyi). (**A**) Genes repressed upon Hey targeting in ECs, and (**A'**) genes activated upon Hey targeting in ECs. (**B, C**) Cytoscape-based gene-ontology analyses for genes that are downregulated or upregulated, respectively, upon Hey depletion in ECs (see *Supplementary file 2*). (**D–G'**) Targeting Hey in ECs resulted in increased expression of Hey-putative direct target

*Figure 3 continued on next page*

*Figure 3 continued*

RecQ4 protein in small cells that are LamC[(-)] Delta[(+)] (compare D to E, and F to G). (G') Quantification of polyploid cells (PPCs) that co-express Delta and RecQ4 in control and midguts where Hey is targeted in ECs. Arrows indicate cells shown in insets. Scale bar is 10 μm. (H) Venn diagram comparison of genes regulated by Hey in progenitors (P) and ECs. Gene loci also bound by Hey, as identified by DamID, are indicated by a gray rectangle. Vertical bars indicate the percent of genes whose expression is repressed (green) or activated (red) upon Hey knockdown (I) A non-canonical HES binding site is enriched in Hey bound loci. (J–M) Single gene examples of genomic loci bound by Hey using DamID. Binding profile of Hey over the genomic regions encoding for *delta* (J), *Pdm1* (K), *LamC* (L) *LamDm0* (M). Single peaks and regions are shown in orange and blue, respectively. The mean signal was smoothed +whiskers. Predicted enhancers were identified using modENCODE are depicted as orange lines, and coding regions are shown as black arrowed lines. modencode.org/publications/integrative fly (*Gerstein et al., 2010*); GSE22069 (*Filion et al., 2010*),_(*Roy et al., 2010*).
DOI: https://doi.org/10.7554/eLife.44745.012

The following source data and figure supplements are available for figure 3:

**Source data 1.** Quantification data related to *Figure 3G'* - number of PPC Dl(+) RecQ4(+).
DOI: https://doi.org/10.7554/eLife.44745.018
**Figure supplement 1.** Transcriptional regulation of EC identity by Hey.
DOI: https://doi.org/10.7554/eLife.44745.013
**Figure supplement 2.** Hey is required for progenitor identity and differentiation.
DOI: https://doi.org/10.7554/eLife.44745.014
**Figure supplement 3.** MARCAM analysis of Hey mutant clones Confocal images of adult *Drosophila* midgut epithelium derived from the indicated transgenes, DAPI marks DNA, and scale bar is 50 μm.
DOI: https://doi.org/10.7554/eLife.44745.015
**Figure supplement 3—source data 1.** Raw data for Quantification of PPC that are positive for both GFP and PDM in experimental setting similar to (E and F).
DOI: https://doi.org/10.7554/eLife.44745.016
**Figure supplement 4.** Single gene examples of genomic loci bound by Hey, and PCA analysis of Hey- dependent gene expression experiments.
DOI: https://doi.org/10.7554/eLife.44745.017

To identify putative direct transcriptional effects of Hey, we mapped the genomic loci bound by Hey using DamID chromatin profiling (*Figure 3H–M*, *Figure 3—figure supplement 4A–C*; *Supplementary file 3*; *Greil et al., 2006*; *Orian, 2006*; *Rincon-Arano et al., 2012*). We were unable to generate Dam-Hey transgenic lines, preventing the mapping of Hey-DamID in gut cells. Instead, we performed DamID mapping in *Drosophila* Kc167c cells. Overall, we identified ~4000 Hey-associated genomic regions (*Supplementary file 3*). Hey-bound loci were enriched for non-canonical HES/ Hey-related binding motif, which was also shown to be a preferred binding motif for mammalian Hey proteins (CANNTG, e = $5.4 \times 10^{-4}$, *Figure 3I*; *Heisig et al., 2012*). We found that Hey bound the genomic regions of 32% (127/370) and 35% (127/345) of Hey-regulated genes in ECs and progenitors, respectively (*Figure 3H*). By comparing Hey-DamID with Hey-regulated genes, we identified distinct sets of putative Hey targets in progenitors and ECs with only 10 shared targets, further supporting cell-specific functions of Hey. Moreover, in each group, about ~70% of Hey-regulated genes that were bound by Hey in DamID required Hey for their expression. (*Figure 3H*, *Supplementary files 1,2,4*). For example, Hey binding was observed in predicted enhancer regions of *delta*, *pdm1*, *lamC*, *lamDm0*, *broad*, *γ-tubulin*, and *pointed*, th only residual expression in ISCs a their expression in ECs was shown to be regulated by Hey (*Figure 3J–M*, *Figure 3—figure supplement 4A–C*). A detailed analysis of DamID results will be presented elsewhere.

## Hey regulates chromatin and nuclear organization of ECs

At the chromatin level, Hey binding correlated with genomic regions enriched for histone tail modifications associated with gene activation, such as the dual histone marks H3K4me1[+]/H3K27ac[+], H3K4me2, H3K4me3, H4K16ac (*Figure 4A*, *Figure 4—figure supplement 1A–E*). In contrast, Hey binding was not correlated with histone marks associated with poised enhancers (H3K4me1[+]/ H3K27ac[-]), or silenced chromatin (*Figure 4A*; *Figure 4—figure supplement 1F–H*; modENCODE, 2010; *Ernst et al., 2011*, and reviewed in *Smith and Shilatifard, 2014*). Indeed, we found that Hey regulates histone tail modifications associated with gene activation in vivo. In this set of experiments, we used the regional 103–555 Gal4 driver, which is active only in enterocytes within a sub-gut region, while neighboring ECs outside the targeted region served as control (*Figure 4B Figure 4— figure supplement 2A–E*). Regional downregulation of Hey reduced the H3K27ac signal only in ECs

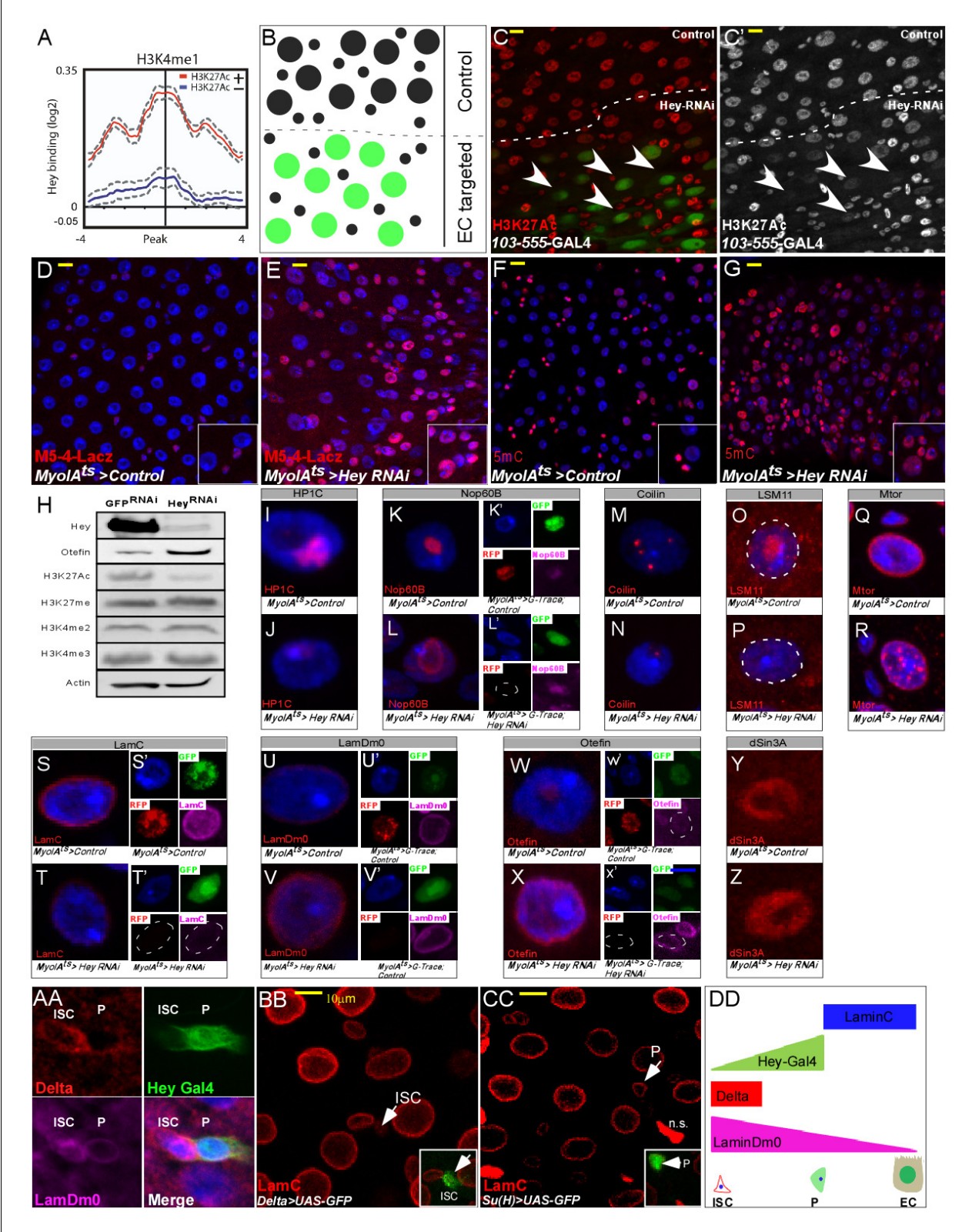

**Figure 4.** Hey regulates chromatin and large-scale nuclear organization. (**A**) End analysis of Hey DamID signal to H3K4me1-enriched regions containing H3K27Ac (red). (**B**) Schematic diagram of regional EC-targeting system. GFP marks ECs where RNAi is active (targeted region). (**C, C′**) Confocal images of regional targeting of Hey in ECs. H3K27Ac is shown in red, targeted ECs are GFP-positive, and the targeted region is below the dashed white line. (**D–G**) RNAi-mediated targeting of Hey using *Myo1A*-GAL4ts for 48 hr resulted in increased and ectopic expression of the *esg* gene stem cell enhancer

*Figure 4 continued on next page*

*Figure 4 continued*

(*M5-4*::LacZ) (**D, E**), and in an increase in chromatin accessibility (**F, G**) as measured by *M*-SSL-I assay, 5mC is shown in red (see text for details). DAPI (blue) marks DNA, and scale bar is 10 μm. (**H**) Western blot analyses of the indicated proteins in extract derived from S2R + cells that were transfected with GFP RNAi or Hey RNAi. Actin serves as loading control. (**I–Z**). Single cell images of RNAi-mediated reduction of Hey, but not control, in ECs resulted in changes in the expression and distribution of the indicated proteins associated with intranuclear organelles and chromatin/nuclear domains; Representative images of G-TRACE analysis are shown in (**K', L', S' T', U' V', W', X'**). DAPI marks DNA. (**AA-DD**) expression of nuclear lamins in midgut cells (**AA**) ISCs are Delta positive (red) and express LamDm0 (purple). EBs are marked by *Hey*-Gal4 >GFP (green). (**BB, CC**) Immunostaining of LamC (red) and the indicated cell-specific transgenes (green), arrows indicate cells in insets: (**BB**) ISCs are marked by Delta-GAL4 >UAS GFP; (**CC**) EBs are marked by Su(H)-GAL4 > UAS GFP. (**DD**) Distribution of lamins during ISC differentiation to ECs.
DOI: https://doi.org/10.7554/eLife.44745.019

The following source data and figure supplements are available for figure 4:

**Figure supplement 1.** Correlations between Hey-DamID-bound loci and histone tail modifications.
DOI: https://doi.org/10.7554/eLife.44745.020

**Figure supplement 2.** Characterization of regional targeting system (**A–E**) Regional EC targeting line.
DOI: https://doi.org/10.7554/eLife.44745.021

**Figure supplement 3.** Characterization of the *esg*-M5-4 lacZ enhancer expression in midgut cells, and quantification of the effect of loss of Hey on large scale chromatin and nuclear organization (**A–D**) Confocal images of adult *Drosophila* midgut epithelium derived from the indicated transgenes, DAPI marks DNA, and scale bar is 50 μm.
DOI: https://doi.org/10.7554/eLife.44745.022

**Figure supplement 3—source data 1.** Quantification data related to *Figure 4—figure supplement 3*.
DOI: https://doi.org/10.7554/eLife.44745.023

located within the targeted zone (*Figure 4C and C'* and quantified in *Figure 5—figure supplement 1C*). Similar results were also observed upon inactivating Hey using the general EC MyoIA$^{ts}$Gal4 system (*Figure 4—figure supplement 3H–K*). Furthermore, H3K27ac protein levels were reduced in *Drosophila* Schneider S2R cells transduced with Hey RNAi (*Figure 4H*), suggesting that Hey may be required to maintain H3K27ac. Alongside these observations in ECs, Hey represses the expression of the stem cell *M5-4* enhancer sequence, which promotes the expression of the stem cell-related *esg* TF (*Gönczy and DiNardo, 1996*). We observed that, in control guts, the *M5-4* enhancer drives Lac-Z expression only in progenitor cells but not in ECs or EEs (*Figure 4D*, *Figure 4—figure supplement 3A–D*). Following loss of Hey in ECs, however, the *M5-4* enhancer drove robust lac-Z expression not only in stem cells but also in polyploid cells and specifically in PPC** (*Figure 4E*, *Figure 4—figure supplement 3E,F and G*-TRACE analysis 2K, 2L). This suggests that under physiological conditions and in ECs, Hey represses the activity of a progenitor-specific enhancer.

We hypothesized that the ectopic expression of a stem-cell enhancer in PPCs** as well as PPCs[*] may be due to permissive changes in chromatin organization. We therefore investigated whether loss of Hey in ECs results in a global change in chromatin conformation using an M. SssI methylation-based chromatin accessibility assay (*Figure 4F,G*; *Rincon-Arano et al., 2012*). In brief, the M. SssI enzyme efficiently methylates CpG dinucleotides in vitro depending on chromatin accessibility. This methylation is endogenously minimal in differentiated somatic *Drosophila* cells. The methylated dinucleotides are subsequently detected using 5mCmAb (*Bell et al., 2010*). Only minimal 5mC methylation was detected in control gut ECs (9%, n = 206), but upon Hey knockdown, a significant increase in 5mC was detected in PPCs using immunofluorescence with the 5mC antibody likely (22% n = 406; p<0.001) (*Figure 4F,G* and quantitated in *Figure 4—figure supplement 3G*).

While Hey proteins are sequence-specific transcription factors that regulate the expression of distinct genes, these observations led us to test whether loss of Hey affected global organization of differentiated EC nuclei. We therefore examined the expression and localization of proteins that are associated with chromatin regions and non-chromatin subnuclear organelles (*Figure 4I–Z* and qauntified in *Figure 4—figure supplement 3L*). RNAi-mediated depletion of Hey resulted in impaired expression and distribution of heterochromatin protein 1 c (HP1c), which was no longer localized to the chromocenter (*Figure 4I,J*). It also resulted in an increased distribution of Nop60B and Coilin, which mark the nucleolus and Cajal bodies, respectively (*Figure 4K–N*). We also observed impairment in the nuclear localization of LSM11 that is associated with the histone locus bodies (HLBs) (*Figure 4O,P*). At the nuclear periphery, localization of Mtor, a nuclear envelope component, was distorted. Alongside we observed changes in the expression of nuclear lamins (*Figure 4Q–V*).

We observed reduced expression of Lamin C and increased expression of LamDm0. We also observed increased expression of the LamDm0 binding protein, Otefin, as well as its redistribution to the nuclear periphery (*Figure 4W–X' and H*, and qauntified in *Figure 4—figure supplement 3*). In contrast, the immunofluorescence signal of the co-repressor dSin3A, which functions in HES-mediated repression, was identical in both Hey-RNAi- and control-targeted gut (*Figure 4Y,Z*, *Figure 1—figure supplement 3K,L*). Thus, loss of Hey results in global changes in the organization of the EC nucleus.

## Hey regulates the expression of nuclear lamins

Maintaining cell identity requires the establishment of high-order nuclear organization involving nuclear lamins (*Kohwi et al., 2013*; *Harr et al., 2015*). Our binding and expression data predicted that Hey partially shapes the organization of the differentiated EC nucleus by regulating lamin expression (*Figures 3L, M* and *4S–V'*, *Supplementary files 1*,*3*). We therefore further examined the regulation of nuclear lamins by Hey in ECs and the role of lamins in supervising EC identity.

Nuclear lamins are essential for establishing facultative heterochromatin and gene silencing (*Dialynas et al., 2010*; *Kohwi et al., 2013*; *Gruenbaum and Foisner, 2015*). They contribute, in part, to the specification of lamin-associated domain regions (LADs) characterized by low gene expression levels (*Guelen et al., 2008*; *Barton et al., 2015*). The *D. melanogaster* genome encodes for two intermediate filament lamin genes: LaminDm0 (LamDm0), a type-B lamin, and lamin C (LamC), a type-A lamin expressed predominantly in differentiated cells.

Remarkably, the distribution of A and B type nuclear lamins is distinctive: LamDm0 is highly expressed in both ISCs and EEs, but is present in only low levels in EBs and ECs. In contrast, LamC is expressed in EBs and ECs with only residual expression in ISCs and EEs (*Figure 4AA-DD*, *Figure 5—figure supplement 1D,E*).

We found that Hey establishes and maintains a unique organization of lamins in ECs. Regional depletion, as well as general depletion of Hey in ECs resulted in a decline in LamC expression and in an increased LamDm0 protein levels in ECs (*Figure 5A–B'*, *Figure 5—figure supplement 1A–C and F–I*). Using G-TRACE analyses, we also observed that PPCs[**] exhibits ectopic expression of LamDm0 (*Figure 5C,D*) and reduced levels of LamC protein (*Figure 5E,F*). Importantly, the changes in LamC and LamDm0 expression, as well as nuclear organization (such as the localization of Mtor), were not observed upon exposure to 5 mM paraquat, that induces rapid progenitor proliferation (*Figure 5—figure supplement 2I-P*; *Chatterjee and Ip, 2009*). Thus, these changes are directly related to loss of Hey supervision in ECs.

Moreover, forced expression of Hey in progenitor cells (esg-GPF positive cells) resulted in reduced expression of the stem cell-related lamin, LamDm0, as well as Delta (*Figure 5G–I*). Only 4.5% of progenitor cells co-express LamDm0 and Delta compare to 100% in control (*Figure 5I*). Loss of Pdm1 in ECs similarly resulted in an inability to maintain the expression of EC-specific genes like LamC and Caudal and in the upregulation of LamDm0 (*Figure 5—figure supplement 2C–H*). Unlike the case of Hey, however, RNA-mediated knockdown of Pdm1 in ECs did not result in ectopic expression of Delta in these cells (*Figure 5—figure supplement 2A,B*). Moreover, co-expression of Pdm1 along with Hey elimination in ECs did not suppress Hey phenotypes (not shown), suggesting that, in ECs, Pdm1 function is not redundant with Hey.

## Nuclear lamins determine cell identity

It is commonly thought that the association of nuclear lamins with the genome is inhibitory (*Barton et al., 2015*; *Gruenbaum and Foisner, 2015*). We hypothesized that lamins maintain cell identity in part by inhibiting the expression of non-relevant transcriptional programs. Indeed, eliminating LamC in ECs using LamC-RNAi resulted in ectopic expression of the stem cell marker Delta on the surface of PPCs (*Figure 5J,K,T*). Eliminating LamDm0 from progenitor cells similarly resulted in ectopic expression of the founder EC gene Pdm1 (*Figure 5L,M,U*). Moreover, conditional ectopic expression of the differentiated LamC, but not of LamDm0, in progenitor cells resulted in a decline in the expression of the stem cell marker Delta (*Figure 5N,O,V*, and controls shown in *Figure 5—figure supplement 1N,O*). Expression of stem cell-related LamDm0 in ECs, but not of LamC, resulted in reduced expression of the EC-related transcription factors, Pdm1 and Odd-skipped (*Figure 5P,Q,W*; *Figure 5—figure supplement 2J–M*).

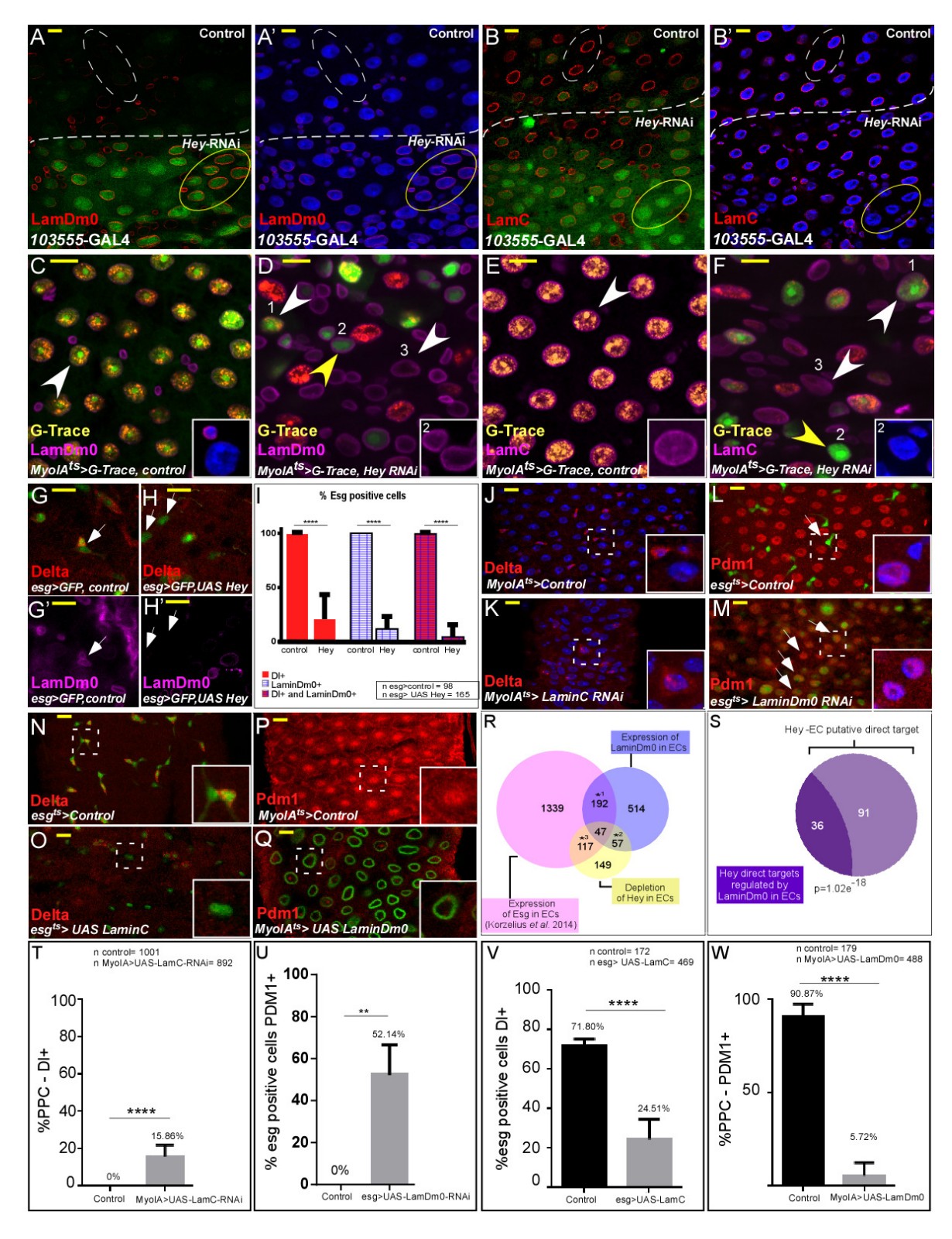

**Figure 5.** Hey regulates the expression of nuclear lamins, which determine cell identity. (**A–B'**) Effects of regional Hey targeting in ECs on the expression of LamDm0 and LamC. Hey RNAi is expressed in ECs that co-express GFP and are below the dashed line. RNAi activation was for 48 hr. (**C–F**) G-TRACE analyses of LamDm0 (**C, D**) and LamC (**E, F**) in control and Hey-targeted ECs. Lamin proteins are shown in purple. Numbers in each figure indicate: (1) EC; (2) PPC**; (3). PPC*. Yellow arrows indicate PPC** shown in insets. (**G–H'**) Expression of Hey (**H, H'**), but not control (**G, G'**), in

*Figure 5 continued on next page*

*Figure 5 continued*

progenitors cells reduces the expression of LamDm0 and Delta. (I) Quantification of a large midgut area in similar setting to G-H' (****=P < 0.0001). (J–Q) Confocal images from guts derived from the indicated transgenes and antibodies. (J, K) RNAi-dependent knockdown of LamC in ECs using *MyoIA*-Gal4/Gal80$^{ts}$ results in ectopic expression of Delta (red). (L, M) RNAi-mediated targeting of LamDm0 in progenitor cells using *esg*-Gal4/Gal80$^{ts}$ (along with UAS-GFP) results in ectopic Pdm1 expression in these cells (GFP$^+$, red). (N, O) Ectopic expression of LamC in progenitor cells results in reduced Delta expression (red). White dashed squares indicate cells shown in insets. (P, Q) Ectopic expression in ECs of LamDm0-GFP (Q, green), but not of control, (P) resulted in reduced Pdm1 expression (red). (R) Venn diagram comparison of the indicated differentially expressed genes (DEG). *1, p=4.03e$^{-72}$; *2, p=3.61e$^{-79}$; *3, p=1.35e$^{-85}$ (S) Comparison between Hey-putative direct targets in ECs and genes regulated by ectopic expression of LamDm0-GFP in ECs. (T–W) Quantification biological repeat similar to the ones shown in J-Q. (T) PPCs expressing Delta. (U) Cells that are *esg*::GFP positive and that are positive for Pdm1. (V) Cells that are *esg*::GFP positive and are positive for Delta. (W) PPC positive for Pdm1 in experiments similar to the ones shown in J-Q respectively. ****=p < 0.0001.

DOI: https://doi.org/10.7554/eLife.44745.024

The following source data and figure supplements are available for figure 5:

**Source data 1.** Quantification data related to *Figure 5*.
DOI: https://doi.org/10.7554/eLife.44745.030

**Figure supplement 1.** Hey and nuclear lamins regulate cell identity.
DOI: https://doi.org/10.7554/eLife.44745.025

**Figure supplement 1—source data 1.** Quantification data number of PPC GFP(+) cells.
DOI: https://doi.org/10.7554/eLife.44745.026

**Figure supplement 2.** The effects on EC identity upon RNAi-dependent targeting of Pdm1, or Paraquat treatment.
DOI: https://doi.org/10.7554/eLife.44745.027

**Figure supplement 3.** Transcriptional analysis of regulation of cell identity by LamDm0.
DOI: https://doi.org/10.7554/eLife.44745.028

**Figure supplement 4.** PCA analysis of LamDm0-dependent gene expression experiments.
DOI: https://doi.org/10.7554/eLife.44745.029

To gain a comprehensive view on the program(s) regulated by the ectopic over-expression of LamDm0 in ECs, we performed RNA-Seq analysis. Using MyoIA$^{ts}$-GAL4, driving the expression of UAS-LamDm0 we identified 810 DEGs upon LamDm0 expression in ECs compared with control (UAS-GFP, PCA analysis shown in *Figure 5—figure supplement 4*, and see Materials and methods). We identified 155 genes (FDR < 0.05), termed 'Group 1' (G1), that are highly expressed in ECs and whose expression was repressed by LamDm0. These genes encode for proteins that are involved in differentiated ECs physiology and metabolism (*Figure 5—figure supplement 4A,C,E*). We also identified 154 genes (FDR < 0.05), termed 'Group2' (G2), that exhibit upregulated expression upon expression of LamDm0 and are highly expressed in progenitor cells, but not in ECs. These genes are involved in cell cycle, DNA repair, chromatin organization and transcription, RNA transport, and lateral inhibition (*Figure 5—figure supplement 4B,D*). Data comparison showed that many LamDm0 DEGs are co-regulated by Hey and are repressed by the expression of the progenitor transcription factor Escargot (Esg) in ECs (*Figure 5R*; *Korzelius et al., 2014*) and in comparison to fly gut gene expression resource *Figure 3—figure supplement 3E*; https://flygut.epfl.ch/expressions). Moreover, 30% (36/127) of Hey putative direct targets are regulated by LamDm0 in ECs (*Figure 5S*). Thus, loss of EC identity stems from active reconfiguration of EC nuclei, as a result of the acute loss of Hey or aberrant lamin expression.

## Perturbation of hey or lamin expression in ECs results in impaired gut homeostasis and reduced organismal survival

RNA-mediated knockdown of Hey in ECs not only impaired EC identity but also affected the entire epithelial tissue and gut homeostasis activating stress/regeneration pathways and impaired tissue integrity (*Figure 1K–N*, and *Figure 6*). EC-specific Hey knockdown disrupted the organization of the epithelial tissue, as shown by the decline in expression of EC-specific cell adhesion proteins MESH, Snakeskin (SSK), Fasciclin (FasIII), and the septate junction protein Dlg (*Figures 6A,B* and *2I, I'*; *Figure 6—figure supplement 1A–F*; *Izumi et al., 2012*; *Yanagihashi et al., 2012*; *Korzelius et al., 2014*). Concomitantly, we observed the ectopic expression of Armadillo protein (*Drosophila* β-catenin) on the surface of polyploid cells (*Figure 6C,D*).

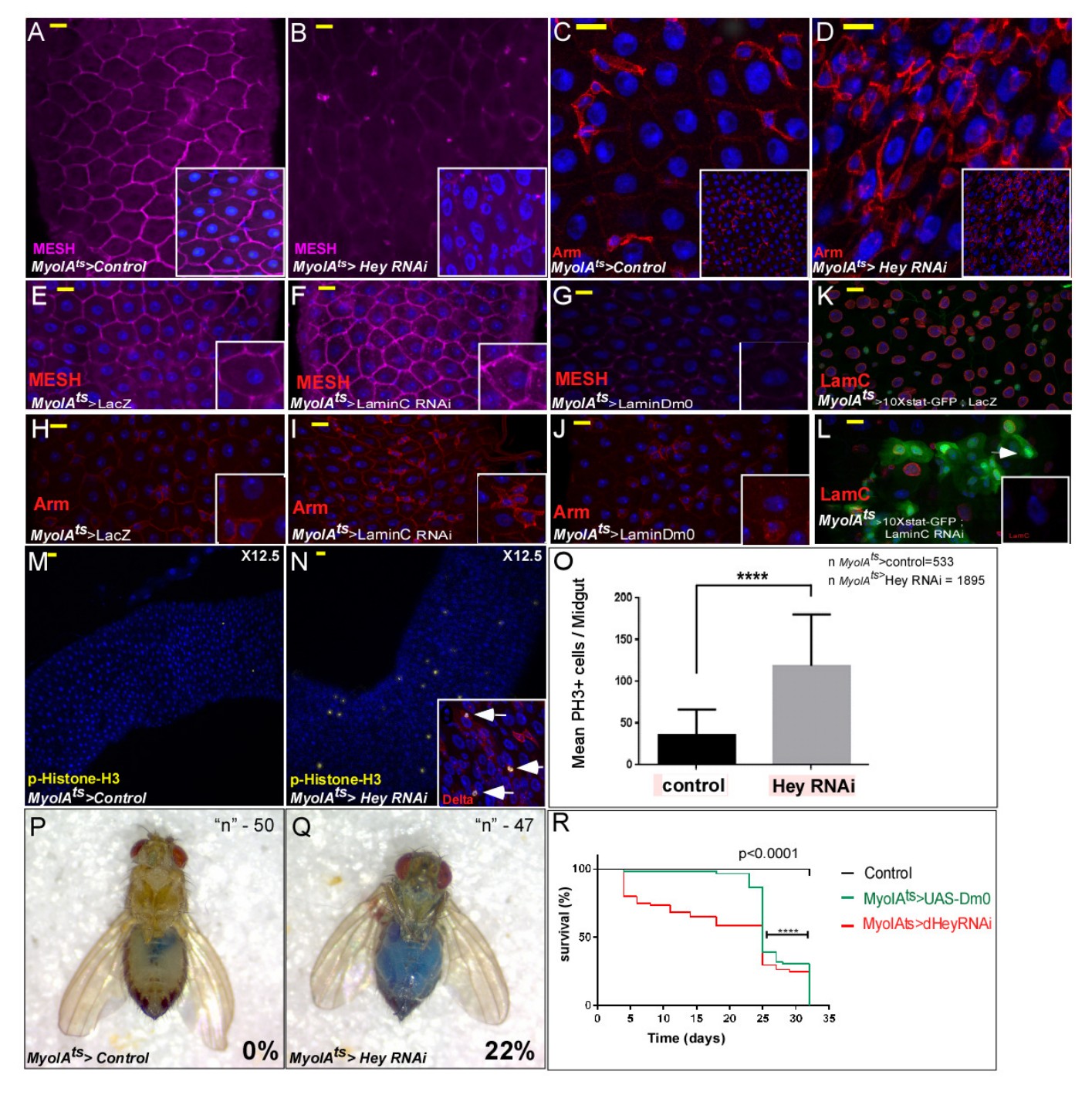

**Figure 6.** Perturbation in the expression of identity supervisors impairs gut homeostasis and gut integrity and reduces survival. (A–N) Confocal images of adult *Drosophila* midgut epithelium derived from the indicated transgenes. Activation of UAS-transgenic lines was for 48 hr. DAPI marks DNA, and scale bar is 10 µm. (A–D) Loss of Hey in ECs results in reduced expression of the tight junction protein MESH at cell boundaries (A, B), and ectopic Armadillo protein levels (Arm, red) on the surface of large polyploid cells (C, D). Insets in A-D are smaller magnifications of the epithelial tissue. (E–J) Expression of MESH (E–G) or Arm (H–J) in midguts derived from the indicated transgenic lines expressed in ECs under the control of MyoIA-Gal4/ Gal80ts. (K–L). Expression of STAT reporter transgenic lines in control (K), or UAS-LamC-RNAi (L), expressed in ECs using MyoIA-Gal4/Gal80ts. (M–O) Targeting Hey results in an increased number of small cells that are positive for both pH3 and Delta. (O) Quantification of the number of phospho-H3 cell positive in 31 midguts (15 control and 16 of Hey-i) in similar experimental setting to the ones shown in M and N. (P, Q) Targeting Hey in ECs results in leakage of blue-colored food into the abdomen, where 22% of Hey-RNAi flies show loss of gut integrity versus 0% in control flies (n = 94, p<0.001). (R) Survival analyses of flies expressing the indicated transgenes in ECs under the control of MyoIA-Gal4/Gal80ts (n = 180, ****p<0.0001).

*Figure 6 continued on next page*

*Figure 6 continued*

DOI: https://doi.org/10.7554/eLife.44745.031

The following source data and figure supplement are available for figure 6:

**Source data 1.** Quantification data related to *Figure 6O* - total number of phospho H3(+) cells per ROI.

DOI: https://doi.org/10.7554/eLife.44745.033

**Figure supplement 1.** Loss of Hey impairs gut homeostasis and gut integrity.

DOI: https://doi.org/10.7554/eLife.44745.032

Perturbation in Lamin expression also effected gut integrity and homeostasis. Expression of LamDm0 in ECs repressed the expression of EC-specific genes, as seen for example in the reduced expression of adhesion protein MESH as opposed to the RNAi-dependent elimination of LamC in ECs, which had no effect on MESH expression (*Figure 6E–G*). Likewise, Arm, which is expressed only on the surface of progenitor cells, was ectopically expressed on the surface of PPCs upon LamC elimination, but not upon LamDm0 over-expression in ECs (*Figure 6H–J*). Eliminating LamC similarly resulted in ectopic expression of STAT-GFP reporter in polyploid cells (*Figure 6K–L*). Taken together, we conclude that lamins determine cell identity and have non-overlapping cell-specific repressive functions. The continued expression of LamC in ECs is required in order to prevent the expression of progenitor genes but not of differentiated EC genes.

Upon targeting Hey or LamC in ECs, we also observed an increase in mitosis of small cells, potentially ISCs (positive for both Delta and the mitotic marker phosphorylated histone H3) (*Figure 6M,N* quantified in 6O, and not shown), culminating in a loss of overall gut integrity, as exemplified by the leaking of blue-colored food from the gut lumen into the abdominal cavity (*Figure 6P,Q*; *Rera et al., 2011*).

At the organismal level, Hey knockdown or expression of LamDm0 in ECs shortened the fly's life-span. Only 2% of targeted adults survived past 32 days compared with 98% of control flies (*Figure 6R*; $DT_{50}$23 days, n = 300, p<0.0001). Thus, continued Hey expression in ECs and the precise expression of lamins are essential for proper gut homeostasis, tissue integrity, and adult survival.

## Aging impairs EC identity, which is preserved by continued Hey expression in aged ECs

Perturbation in nuclear organization is a hallmark of aged cells and tissues. In the aging *Drosophila* immune system, a decline in the type-B lamin, LamDm0, in fat body cells induces systemic inflammation including in the mid-gut (*Chen et al., 2014*). Aging also resulted in loss of intestinal compartmentalization, and microbiota-dysbiosis, all leading to reduces life-span (*Li et al., 2016*). During aging, EC physiology, midgut integrity, and homeostasis are compromised (reviewed in *Miguel-Aliaga et al., 2018*). Indeed, we noticed that Hey protein levels in ECs decline with age (*Figure 7A, B*). Moreover, the phenotypes resulting from acute conditional knockdown of Hey in young flies (2–4 days old) are highly similar to the phenotypes observed in wildtype guts derived from aged flies (3–4 weeks old). As seen in *Figure 7C–J*, aging ECs exhibit a reduction in the MyoIA-GFP signal, a decline in Pdm1 and LamC expression, along with ectopic expression of Delta and LamDm0 and overall reduced gut integrity (*Figure 7C, E, G, I and K*). Indeed, continued expression of Hey in aged ECs using MyoIA^ts-Gal4 prevented the ectopic expression of Delta and LamDm0 (*Figure 7D and J*), restored the expression of Pdm1 (*Figure 7E and F*), of LamC (*Figure 7G and H*), and of overall EC morphology, compared with control. Finally, expression of Hey prevented loss of gut integrity as revealed by the 'smurf' assay (*Figure 7K and L*). Thus, by supervising EC identity, Hey attenuates epithelial aging and safeguards gut integrity.

## Discussion

We identified Hey, but not other HES transcription factors, as a critical supervisor of EC identity and suggest the following working model (*Figure 7M*): Hey regulates EC identity in part by establishing and sustaining a transcriptional switch in the expression of nuclear lamins. In ISCs, the dominant lamin is LamDm0, which prevents the expression of differentiated genes. During differentiation and in differentiated ECs, Hey represses the expression of LamDm0, enabling the expression of genes

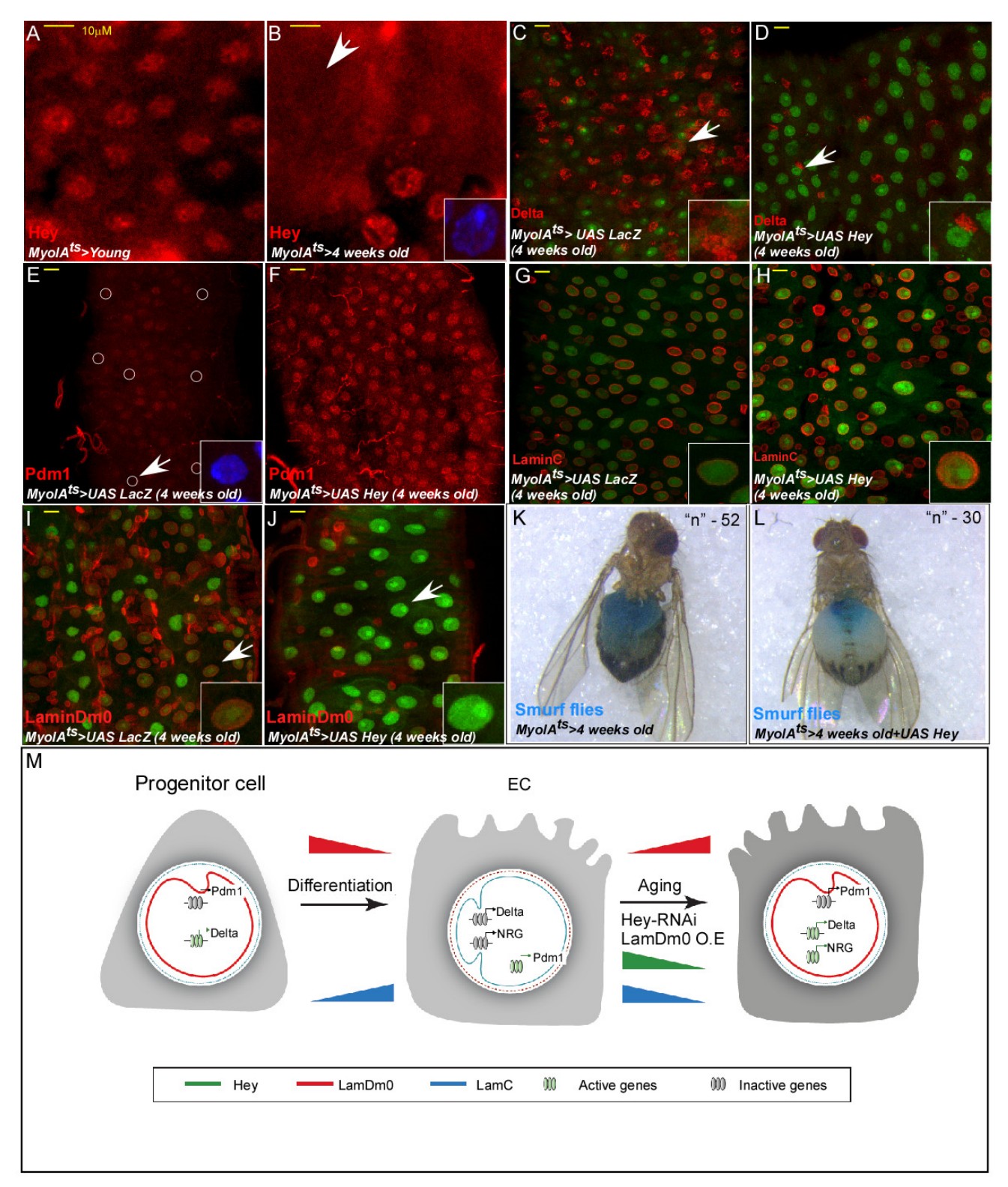

**Figure 7.** Hey protein level declines in aged ECs and its over-expression suppresses aging phenotypes. (**A, B**) Confocal images of Hey protein (red) in young (**A**) and aged (**B**) adult midgut epithelium. DAPI marks DNA, and scale bar is 10 µm. (**C–J**) Confocal images with the indicated antibodies of ECs expressing either UAS-LacZ (control, (**C, E, G, I**), or UAS-Hey (**D, F, H, J**) transgenes using the MyoIA[ts]-Gal4. White arrow indicates cells shown in insets. (**K, L**) Aged ECs exhibit leakage of blue-colored food into the abdomen, which is prevented by expression of UAS-Hey (**L**), but not control (**K**) where

*Figure 7 continued on next page*

*Figure 7 continued*

only 7% of aged flies expressing Hey show loss of gut integrity versus 28% in control aged flies (n = 52, and 30 respectively, p<0.05). (**M**) A model for regulation of cell identity by Hey and nuclear lamins; NRG, non-relevant genes (see text for details).

DOI: https://doi.org/10.7554/eLife.44745.034

required for EC physiology and function. In addition, Hey promotes the expression of EC gene signatures, including Pdm1 and LamC, the latter inhibits the expression of stem cell- and non-gut-related genes in ECs. Hey loss during aging or upon its acute genetic ablation in young midguts, results in ectopic expression of LamDm0 and subsequently silencing of EC programs including critical EC TFs (e.g. Pdm1 and Odd-skipped). Concomitantly, Hey loss in ECs causes a decline in LamC and, as a consequence, ectopic expression of stem cell- and non-gut-related genes that are normally repressed by LamC in ECs.

## Transcriptional regulation of EC identity by Hey

Although HES/Hey proteins are well-known repressors (*Heisig et al., 2012*), our study suggests that in ECs, Hey has both repressive and activating functions. This dual transcriptional activity of Hey in *Drosophila* is similar to its classification in mammalian transcription factors (*Stampfel et al., 2015*) and is important for maintaining EC identity. In ECs, Hey represses the expression of numerous progenitor-related genes whose expression depends on the progenitor transcription factor Esg, such as *Delta*, *snail*, *piwi*, and the ISC-related lamin, LamDm0 (*Dutta et al., 2015*; *Korzelius et al., 2014*). Hey concomitantly drives the expression of key EC founder genes (e.g. *pdm1, odd)* and EC-differentiated genes including LamC. Of specific interest is the regulation of the EC founder *Pdm1* (*Korzelius et al., 2014*). We found that *pdm1* mRNA levels increase during progenitor-to-EC differentiation and that Hey safeguards *pdm1* expression through two mechanisms. First, Hey binds to a predicted enhancer within the *pdm1* gene, inducing and maintaining Pdm1 expression in ECs. Second, Hey-dependent repression of *lamDm0* indirectly ensures continuous *pdm1* expression. Collectively these data place Hey upstream of *pdm*1 and nuclear lamins. Pdm1 expression in Hey-targeted ECs was not, however, sufficient to restore EC identity or prevent the expression of a stem cell gene like *Delta*, suggesting that Hey functions are not compensated by Pdm1 expression alone. Yet, our observation that both Hey and Pdm1 regulate genes like *LamDm0* warrants an investigation of the nature of the interaction between the two proteins in ECs.

## Hey regulates chromatin and nuclear organization

While Hey is a sequence-specific transcription factor, Hey loss affects global nuclear organization, resulting in increased chromatin accessibility as well as changes in the expression and distribution of protein that are distinctive for intranuclear organelles and nuclear domains.

Interestingly, the phenotypes associated with loss of Hey in ECs are reminiscent of the phenotype observed upon loss of Wiskott-Aldrich syndrome protein (WASP), which binds directly to LamDm0 (*Verboon et al., 2015*). This may indicate that large-scale nuclear organization involves a general network that includes nuclear lamins. One possibility is that this regulation may also be dependent on enhancer-associated RNAs (eRNAs), or other non-coding RNA molecules (*Beagrie and Pombo, 2016*; *Brazão et al., 2016*). Future challenges are therefore to identify the molecular mechanisms that determine the differential effect of Hey on enhancers within the same cell, and the exact mechanisms by which Hey maintains subnuclear organization.

Recent work on reprogramming and differentiation distinguished between transcription factors that possess genome-shaping abilities (pioneer factors) and terminal transcription factors, which together shape the transcription network required to establish cell-specific states (*Iwafuchi-Doi and Zaret, 2016*; *Morris, 2016*). We suggest that Hey is likely part of a new class of factors characterized by dual transcriptional activity with profound impact on chromatin and nuclear organization, and that it is situated below pioneer factors but upstream to terminal factors (e.g. Pdm1).

## Nuclear lamins determine EC identity

A key aspect in maintaining large-scale nuclear organization and cell identity is mediated by nuclear lamins. In general, A-type and B-type lamins are co-expressed in many tissues within the same cell,

yet form separate filament networks. In the midgut, however, while ISCs and EEs predominantly express the B-type lamin LamDm0, EBs and ECs express the A-type lamin LamC, likely defining a unique nuclear organization in each cell type. Interestingly, a sequential role of lamins in the establishment of facultative heterochromatin was reported in vertebrates. During the development of the mouse retina, early expression of Lamin-B receptor (LBR) is replaced by LamA during later stages of neuronal (rods) differentiation and, accordingly, LBR and LamA reciprocally regulate the expression of muscle-specific genes during myoblast development (*Solovei et al., 2013*).

Our study established that in each cell type, the dominant lamin is required in order to inhibit the expression of non-relevant programs. In EBs and differentiated ECs, LamC mediates silencing of non-EC genes. This is likely a conserved role for A/C-type lamins, as human LamA inhibits the transcription of non-adipocyte programs in differentiated adipocytes by binding in the vicinity of TSS (*Lund et al., 2013*). Indeed, loss of LamC in ECs resulted in ectopic expression of Delta in PPCs[*] as well PPCs[**], exemplifying its role in repressing progenitors-related genes. Over-expression of LamC in Hey-targeted ECs was not, however, sufficient to repress ectopic Delta expression (data not shown), suggesting that the repressive activity of LamC may require the presence of Hey or a Hey-regulated protein(s) on these gene regions. Furthermore, silencing of stem cell-related genes in EC likely requires other proteins that may likely be polycomb group proteins (PcG), which are involved in anchoring chromatin to the nuclear lamina and in the generation of facultative heterochromatin (*Cesarini et al., 2015*; *Gruenbaum and Foisner, 2015*; *Gonzalez-Sandoval and Gasser, 2016*). As our manuscript was under review Petrovsky and Groβhans also characterized the expression of nuclear lamins in the midgut (*Petrovsky and Großhans, 2018*). Similar to our observations they reported that LamDm0 and Otefin are enriched in progenitor cells. They also reported the expression of LamC in progenitors cells. Our work using ISC and EB specific markers further established that in progenitors high levels of LamC are present in EBs and only minimal amount is expressed in ISCs.

## Cell-specific roles for Hey, gut homeostasis, and the impact of Hey loss during aging

Hey is highly expressed in EBs where it is required for establishing EB identity and promoting Notch-dependent progenitor differentiation to ECs, similar to other HES proteins (*Ohlstein and Spradling, 2006*; *Bardin et al., 2010*; *Perdigoto et al., 2011*). Notch activity is not, however, detected in ECs under normal physiological conditions. The observed Notch-reporter activity in PPCs upon Hey targeting in ECs, may reflect stress-increased ISC proliferation and Notch activation in enlarged progenitors. Alternatively, this ectopic activity may suggest that Hey limits Notch activity in ECs. Moreover, targeting *Notch* or *Su(H)* in ECs had no detectable phenotype. The function of Hey in ECs resembles the Notch-independent function of Hey1, that is maintaining pillar cell fate in the organ of Corti and in endothelial cells (*Doetzlhofer et al., 2009*; *Wöltje et al., 2015*). Furthermore, while Hey is required for EC and EB identity, it is not required in order to maintain EE identity. Unlike ECs, EEs express predominantly LamDm0 and only minimally express LamC. Thus, the differentiated identity of EEs is independent of Hey and is likely maintained by other mechanisms.

The phenotypic changes following Hey knockdown are rapid, last for days, and are only partially reversible. In addition, regional targeting experiments showed that the response to targeting Hey in ECs is observed on progenitors adjacent to, but outside of the targeted region. This non-autonomous progenitor response likely involves diffusible factors secreted from PPCs[**] or other gut cells by ligands that induce ISC self-renewal such as unpaired (UPD1, UPD2). Indeed, upon Hey or LamC loss in ECs we observed enhanced activity of a JAK-STAT pathway reporter in progenitor cells as well as in polyploid cells. The regenerative process may similarly involve additional pathways that regulate stem cell regeneration (*Biteau et al., 2008*; *Choi et al., 2008*; *Foronda et al., 2014*). In the absence of Hey, this regenerative response results in abnormal differentiation of progenitors (PPC*). Mis-differentiation was also observed during direct mammalian cell reprogramming, where reprogrammed cells were incapable of fully differentiating and expressed gene signature remnants of previous fates (*Cahan et al., 2014*).

Our study and other reports suggest that differentiated cells have an inherent ability to reactivate de-differentiation, trans-determination, and in some cases, trans-differentiation programs. This propensity, which appears to be a shared property of cells with rapid turnovers, may have resulted from a lack of evolutionary pressure on these cells, or alternatively developed as an advantageous coping

mechanism under challenging physiological conditions (e.g. stem cell exhaustion, infection). It was recently shown that under severe stress conditions, polyploid enterocytes evolve into functional stem cells in a process termed amitosis (*Lucchetta and Ohlstein, 2017*).

## Cell identity, aging, and pathological plasticity

Loss of identity is a hallmark of aging, and rejuvenating of aged organisms, tissues and cells is a long-standing challenge. The mechanisms involved in regulating the differentiated identity are relevant for premature aging syndromes and aging-related diseases. iPS therapies are currently at the heart of rejuvenation strategies for aged tissues. For example, general, conditional, and transient expression of Oct4, Sox2, Klf4, and c-Myc (OSKM) suppressed cellular and physiological hallmarks of aging and prolonged lifespan in a mouse model of HSPG progeria and premature aging syndrome (*Ocampo et al., 2016*).

Along this line, our study highlights the importance of identity guardians acting within the differentiated cell and preventing aging. We found that changes in aged ECs and in gut tissue are in part preventable and reversible. Thus, the development of strategies supporting the expression and maintaining the functionality of identity supervisors together with gene delivery systems may result in tissue rejuvenation that protects the integrity and physiological properties of both individual cells and the entire tissue.

During *Drosophila* and vertebrate aging, or under various stress conditions, differentiated cells in the gut, pancreas, and tracheal epithelium behave similarly (*Biteau et al., 2010*; *Tata and Rajagopal, 2016*), reflecting a general principle. This loss of identity, however, is intimately associated with pathological plasticity, which is highly relevant to a host of maladies including cancer. Thus, at least in some contexts, supervisors of identity such as Hey establish a barrier to tumorigenesis by regulating the expression of nuclear lamins in the differentiated nucleus. Indeed, mutations in human lamins and lamin-binding proteins are associated with heritable diseases and premature aging syndromes (*Bione et al., 1994*; reviewed in *Butin-Israeli et al., 2012*). One well-studied case is that of human LamC ortholog, LamA, which is mutated in Progeria, a disease associated with premature aging and loss of physiological properties of differentiated cells and tissues (*Hegele, 2003*). However, further studies are needed in order to link loss of identity and premature aging in progeria models and patients, to the consequences of physiological aging. Therefore, the identification of conserved mechanisms underlying the regulation of identity networks in model organisms such as *Drosophila* has broad implications for addressing pathological conditions in humans, and specifically age-related diseases.

## Materials and methods

**Key resources table**

| Reagent type (species) Or resource | Designation | Source or reference | Identifiers | Additional information |
|---|---|---|---|---|
| Genetic reagent (*D. melanogaster*) | w⁻, *Hey*-Gal4, UAS-CD8-GFP/FM7;+;+ | Orian lab | | |
| Genetic reagent (*D. melanogaster*) | pBac{WH}Hey[f06656] | Bloomington | #18997 | |
| Genetic reagent (*D. melanogaster*) | P{neoFRT}42D pwn1 P{Car20y}44B/CyO | Bloomington | #5260 | |
| Genetic reagent (*D. melanogaster*) | yw, hs-flp[122], *Tub-Gal4,Uas-GFP, FRT42D Tub-Gal80/Cyo* | Edgar Bruce lab | | |
| Genetic reagent (*D. melanogaster*) | y, w⁻, hs-flp[122], *Tub-Gal4, UAS GFP/+*; FRT42D *Tub-Gal80*/FRT42D Hey[f06656] | Orian lab | | |

*Continued on next page*

Continued

| Reagent type (species) Or resource | Designation | Source or reference | Identifiers | Additional information |
|---|---|---|---|---|
| Genetic reagent (D. melanogaster) | y, w-, hs-Flp122, Tub-Gal4, UAS GFP/+; FRT42D | Orian lab | | |
| Genetic reagent (D. melanogaster) | w; *MyoIA*-Gal4; *tub*-Gal80ts, UAS-GFP | Edgar Bruce lab | | |
| Genetic reagent (D. melanogaster) | *w*; esg-Gal4, *tub*-Gal80ts UAS-GFP | Edgar Bruce lab | | |
| Genetic reagent (D. melanogaster) | w; Prospero-Gal4 | Edgar Bruce lab | | |
| Genetic reagent (D. melanogaster) | w; *Dl*-Gal4/TM6, Tb | Sarah Bray lab | | |
| Genetic reagent (D. melanogaster) | Su(H)-Gal4 | Sarah Bray lab | | |
| Genetic reagent (D. melanogaster) | Notch-reporter 3.37-gh-LacZ | Sarah Bray lab | | |
| Genetic reagent (D. melanogaster) | 10X-STAT::GFP | Lilach Gilboa lab | | |
| Genetic reagent (D. melanogaster) | M5-4::LacZ | Erika Matunis lab | | |
| Genetic reagent (D. melanogaster) | UAS-LamC | Lori Walworth lab | | |
| Genetic reagent (D. melanogaster) | y* w*; P{w + mW.hs=GawB}NP0203 /CyO,P{w-=UAS lacZ.UW14}UW14 | DGRC | #103555 | |
| Genetic reagent (D. melanogaster) | UAS-Hey-RNAi | VDRC | #30562GD | |
| Genetic reagent (D. melanogaster) | UAS-Hey-RNAi | VDRC | #103570KK | |
| Genetic reagent (D. melanogaster) | UAS-Hey-RNAi | Bloomington | #31898 | |
| Genetic reagent (D. melanogaster) | UAS-Hey-RNAi | Bloomington | #41650 | |
| Genetic reagent (D. melanogaster) | UAS-Hey-RNAi | NIG-FLY | 11194 R-1 | |
| Genetic reagent (D. melanogaster) | UAS-Hey-RNAi | NIG-FLY | 11194 R-3 | |
| Genetic reagent (D. melanogaster) | UAS-LacZ | Bloomington | #1776 | |
| Genetic reagent (D. melanogaster) | UAS-Hairy RNAi | Bloomington | #27738 | |

*Continued on next page*

*Continued*

| Reagent type (species) Or resource | Designation | Source or reference | Identifiers | Additional information |
|---|---|---|---|---|
| Genetic reagent (*D. melanogaster*) | UAS-Her RNAi | Bloomington | #27654 | |
| Genetic reagent (*D. melanogaster*) | UAS-HLHm7 RNAi | Bloomington | #35703, #29327 | |
| Genetic reagent (*D. melanogaster*) | UAS-HLHm5 RNAi | Bloomington | # 26201 | |
| Genetic reagent (*D. melanogaster*) | UAS-CtBP RNAi | Bloomington | #32889 | |
| Genetic reagent (*D. melanogaster*) | UAS-Sir2 RNAi | Bloomington | #32481 | |
| Genetic reagent (*D. melanogaster*) | UAS-LamDm0 RNAi | Bloomington | #31605 | |
| Genetic reagent (*D. melanogaster*) | UAS-LamC RNAi | Bloomington | #31621 | |
| Genetic reagent (*D. melanogaster*) | UAS-LamDm0-GFP | Bloomington | #7376 | |
| Genetic reagent (*D. melanogaster*) | UAS-Pdm1-RNAi | Bloomington | #55305 | |
| Genetic reagent (*D. melanogaster*) | UAS-Notch RNAi | VDRC | #27229GD #100002KK | |
| Genetic reagent (*D. melanogaster*) | UAS-Su(H) RNAi | VDRC | #103597KK | |
| Genetic reagent (*D. melanogaster*) | UAS-Dicer RNAi | VDRC | #24667GD | |
| Genetic reagent (*D. melanogaster*) | UAS-Gro RNAi | VDRC | #6316GD | |
| Genetic reagent (*D. melanogaster*) | 'G-TRACE' (w*; P{UAS-RedStinger}6, P{UAS-FLP.Exel}3, P{Ubi-p63E(FRT.STOP) Stinger}15F2.) | Bloomington | #28281 | |
| Transfected construct (*D. melanogaster* - S2 cells) | pChs-Gal4 vector | | | *Hey* > Gal4 |
| Transfected construct (*D. melanogaster* - S2 cells) | pUASp vector | | | UASp-His-Hey |
| Antibody | Mouse monoclonal IgG1 α-Prospero | DHSB | Prospero (MR1A) | 1:100 |
| Antibody | Mouse monoclonal α-Armadillo | DHSB | N2 7A1 Armadillo | 1:50 |

*Continued on next page*

*Continued*

| Reagent type (species) Or resource | Designation | Source or reference | Identifiers | Additional information |
|---|---|---|---|---|
| Antibody | Mouse monoclonal α-Delta | DHSB | C594.9B | 1:50 |
| Antibody | Mouse monoclonal α 4F3 anti-discs large (Dlg) | DHSB | 4F3 anti-discs | 1:50 |
| Antibody | Mouse monoclonal α−22C10 | DHSB | 22C10 | 1:20 |
| Antibody | Rabbit polyclonal α-HP1 | Susan Purkhurst lab | | 1:1000 |
| Antibody | Mouse monoclonal α-mTor | DHSB | 12F10-5F11 | 1:100 |
| Antibody | Rabbit polyclonal α-βGal | MP Biomedicals | 55976 | 1:500 |
| Antibody | Mouse monoclonal α-Actin | MP Biomedicals | 691001 | (WB) 1:4000 |
| Antibody | Guinea pig α-Hey | *Monastirioti et al. (2010)* | | 1:300 |
| Antibody | Rabbit α-Pdm1 | Dïaz-Benjumea lab | | 1:50 |
| Antibody | Rabbit α-Fasciclin | Mikio Furuse lab | | 1:100 |
| Antibody | Rabbit α-MESH | Mikio Furuse lab | | 1:100 |
| Antibody | Rabbit α-SSK | Mikio Furuse lab | | 1:100 |
| Antibody | Guinea Pig α-caudal | Jeff Reinitz lab | | 1:200 |
| Antibody | Rar/Guinea Pig α-odd-skipped | Jeff Reinitz lab | | 1:100 |
| Antibody | Rabbit polyclonal α-p-histone H3 | Abcam | ab5176 | 1:100 |
| Antibody | Mouse α-Maelstrom | G. Hanon Lab | | 1:50 |
| Antibody | Mouse monoclonal α-γ-Tubulin | Sigma | T5326-200UL | (WB) 1:100 |
| Antibody | Mouse α Lamin C | Yossef Gruenbaum lab | | 1:500 |
| Antibody | Mouse α Otefin | Yossef Gruenbaum lab | | 1:10 |
| Antibody | Rabbit αLamin Dm0 | Yossef Gruenbaum lab | | 1:100 |
| Antibody | Rabbit α RecQ4 | Tao-shih Hsieh lab | | 1:100 |
| Antibody | Rabbit α-H3K4me1 | Ali Shilatifard lab | | 1:100 |
| Antibody | Rabbit α H3K4me2 | Ali Shilatifard lab | | 1:100 |
| Antibody | Rabbit α-H3K4me3 | Ali Shilatifard lab | | 1:100 |
| Antibody | Rabbit α-H3K27me3 | Ali Shilatifard lab | | 1:100 |
| Antibody | Rabbit polyclonal αH3K27Ac | Abcam | ab4729 | 1:100 |
| Antibody | Rabbit polyclonal α-H3K9me3 | Abcam | ab8898 | 1:100 |
| Antibody | Rabbit αNop60B | Steven Pole lab | | 1:100 |
| Antibody | Rabbit α-LSM11 | Joseph Gall lab | | 1:2000 |
| Antibody | Guinee pig α-Coilin | Joseph Gall lab | | 1:2000 |
| Antibody | Alexa Fluor 568 goat anti-mouse IgG1(γ1 | invitrogen | A21124 | 1:1000 |
| Antibody | Alexa Fluor 568 goat anti-mouse IgG (H + L) | invitrogen | A11031 | 1:1000 |
| Antibody | Alexa Fluor 568 goat anti-rabbit IgG (H + L) | invitrogen | A11036 | 1:1000 |
| Antibody | Alexa Fluor 633 goat anti-rabbit IgG (H + L) | invitrogen | A-21070 | 1:1000 |

*Continued on next page*

*Continued*

| Reagent type (species) Or resource | Designation | Source or reference | Identifiers | Additional information |
|---|---|---|---|---|
| Antibody | Alexa Fluor 633 goat anti-mouse IgG1 (γ1) | invitrogen | A-21126 | 1:1000 |
| Antibody | Alexa Fluor 568 goat anti-guinea pig | invitrogen | A11075 | 1:1000 |
| Antibody | Alexa Fluor 633 goat anti-guinea pig | invitrogen | A21105 | 1:1000 |
| Antibody | Alexa Fluor 633 goat anti-rat | invitrogen | A21094 | 1:1000 |
| Antibody | Alexa Fluor 568 goat anti-rat | invitrogen | A11077 | 1:1000 |
| Chemical compound | Diamidino-2-phenylindole* dihydrochl [DAPI] 1 mg | Sigma | D9542-1MG | 1:1000 |
| Chemical compound | Draq-5 | Biostatus | BOS-889–001 R200 | 1:5000 |

- Fly stocks and cell line used in this study
- Antibodies used in this study
- Primers used in this study

Methods:

- Conditional expression of transgenes in specific gut cells
- Conditional G-TRACE analysis
- Clonal analysis using MARCM
- IMARIS-based 3D and quantification
- Paraquat treatment
- Gut dissection and immunofluorescence detection
- Gut integrity and tracing of organismal survival
- Purification and isolation of gut cells
- DamID, expression signatures, gene expression and RNA-seq and bioinformatics analyses.
- RNAi in S2 cells
- 5mC- Chromatin accessibility assay

## Fly stocks used in this study

Fly stocks were maintained on yeast-cornmeal-molasses-malt extract medium at 18°C or as stated in the text. List of lines used is detailed under Supplemental Experimental Procedures. UASp-His-Hey was generated by cloning Hey cDNA with the addition of a 6xHis-tag into the pUASp vector. w⁻, Hey-Gal4;+;+was generated by cloning the 5'UTR proximal genomic region of *Hey* [(-)1450-TSS] into pChs-Gal4 vector. This line was also used to generate the w⁻, Hey-Gal4, UAS-CD8-GFP/FM7;+;+line. Transgenic lines were generated by standard injection protocol using the appropriate primers. w: UASp-Hey; +, was generated by cloning the full cDNA of Hey to pUAS vector (Kpn/Xba) and transgenic lines were generated by injection in our lab.

## Lines used for clonal analysis

We generated FRT-Hey mutant chromosomes by standard recombination on the second chromosome: *pBac{WH}Hey^f06656, P{neoFRT}42D pwn1 P{Car20y}44B/CyO* using Bloomington lines #18997 and #5260. The resulting line was crossed to *yw, hs-flp^122, Tub-Gal4,UAS-GFP, FRT42D Tub-Gal80/Cyo (kind gift of Bruce Edgar)*, and for control we used a FRT 42D on the second. The relevant F1 progeny lines generated–y, w⁻, hs-flp^122, Tub-Gal4, UAS GFP/+; FRT42D *Tub-Gal80*/FRT42D *Hey^f06656* (experiment) and *y, w-, hs-Flp122, Tub-Gal4, UAS GFP/+*; FRT42D (control)– were used for clonal analysis following *Bardin et al. (2010)*.

The following stocks were kind gifts received from various labs, as follows: *w; MyoIA-Gal4; tub-Gal80ts, UAS-GFP; +, w; esg-Gal4, tub-Gal80ts UAS-GFP;+w; Prospero-Gal4* were from Bruce Edgar. *w; Dl-Gal4/TM6, Tb* and *Su(H)-Gal4*, and Notch-reporter 3.37-gh-LacZ were received from

Sarah Bray. *10X-STAT*::GFP and *w; M5-4*::LacZ were received from Lilach Gilboa and Erika Matunis, respectively. UAS-LamC was received from Lori Walworth.

Gal4 for regional EC targeting: *y\* w\**; P{w + mW.hs=GawB}NP0203/CyO,P{w-=UAS lacZ.UW14} UW14 (DGRC #103555). Hey-RNAi knockdown lines: VDRC #30562GD #103570KK, Bloomington #31898, #41650. NIG-FLY: 11194 R-1, 11194 R-3. The following stocks were from Bloomington stock center: UAS-LacZ (#1776); UAS-Hairy RNAi (#27738, 34326); UAS-Her RNAi (#27654); UAS-HLHm7 RNAi (#35703, 29327); UAS-HLHm5 RNAi (# 26201); UAS-CtBP RNAi (#32889); UAS-Sir2 RNAi (#32481), UAS-LamDm0 RNAi (#31605), UAS-LamC RNAi (#31621). UAS-LamDm0-GFP (#7376). UAS-Pdm1-RNAi #55305. The following stocks were from the VDRC stock center: UAS-Notch RNAi (#27229), 100002); UAS-Su(H) RNAi (#103597); UAS-Gro RNAi (#6316). UAS-Dicer RNAi (#24667) 'G-TRACE' transgenic line (# 28281): w\*; P{UAS-RedStinger}6, P{UAS-FLP.Exel}3, P{Ubi-p63E(FRT. STOP)Stinger}15F2.

Cell line: Verified S2 *Drosophila* Schneider cells-DRSC (contributed by N. Perrimon) were obtained from DGRC and by DGRC similar to lot #181 used for modENCODE studies. Cells were gown in S2 Schneider medium supplemented with 10% Fetal calf serum, Glutamine, and Penicillin/Streptomycin antibiotics. S2 cells cultured were grown in 25C0 and routinely tested and found to be mycoplasma-free by PCR.

## Antibodies used in this study

### Primary antibodies

Mouse $\alpha$-Prospero (1:100), mouse $\alpha$-Armadillo (1:50), mouse $\alpha$-Delta (1:50), mouse $\alpha$ 4F3 anti-discs large (Dlg) (1:50), and mouse $\alpha-$22C10 (1:20), and mouse IgG1 mouse IgG1 $\alpha$-mTor (1:100) were all from DHSB; Rabbit $\alpha$-βGal-55976 (1:500) and mouse $\alpha$-Actin (1:4000) were from MP Biomedicals; Rabbit polyclonal $\alpha$-HP1c (1:100) was a gift from Susan Parkhurst. Guinea pig $\alpha$-Hey (1:300, *Monastirioti et al., 2010*) and rabbit $\alpha$-Pdm1 (1:50) were a kind gift from Dïaz-Benjumea; $\alpha$-Fasciclin (1:100), $\alpha$-MESH (1:100), and $\alpha$-SSK (1:100) were a kind gift from Mikio Furuse. $\alpha$-caudal (1:200) and $\alpha$-odd-skipped (1;100) were from Jeff Reinitz; Rabbit $\alpha$-p-histone H3 (1:100, ab5176) was from Abcam; Mouse $\alpha$-Maelstrom (1:50) was a gift from G. Hanon Laboratory; Mouse $\alpha$-$\gamma$-Tubulin (1:100) was from Sigma; Mouse $\alpha$Lamin C (1:500), mouse $\alpha$Otefin (1:10) and rabbit $\alpha$Lamin Dm0 (1:100) were a kind gift from Yossef Gruenbaum; Rabbit $\alpha$RecQ4 (1:100) was a gift from Tao-shih Hsieh. The following histone antibodies were a gift from Ali Shilatifard: Rabbit $\alpha$-H3K4me1 (1:100), rabbit $\alpha$H3K4me2 (1:100), rabbit $\alpha$-H3K4me3 (1:100), and rabbit $\alpha$-H3K27me3 (1:100). Rabbit $\alpha$H3K27Ac (1:100) and rabbit $\alpha$-H3K9me3 (1:100) were from Abcam. Antibodies for were form Rabbit $\alpha$Nop60B (1:100) was from Steven Pole, Rabbit $\alpha$-LSM11 (1:2000), and Guinee pig $\alpha$-Coilin (1:2000) were from Joseph Gall.

### Secondary antibodies

Alexa Fluor 568 goat anti-mouse IgG1($\gamma$1); Alexa Fluor 568 goat anti-mouse IgG (H + L); Alexa Fluor 568 goat anti-rabbit IgG (H + L); Alexa Fluor 633 goat anti-rabbit IgG (H + L); Alexa Fluor 633 goat anti-mouse IgG1 ($\gamma$1); Alexa Fluor 568 goat anti-guinea pig, and Alexa Fluor 633 goat anti-guinea pig (1:1000). Alexa Fluor 633 goat anti-rat, Alexa Fluor 568 goat anti-rat (1:1000). DNA dyes used: Draq5 (1:5000 889–001 R200, Biostatus) and DAPI (1:1000 D9542-1MG, Sigma).

## Primers used in this study

Hey-RNAi 5'-atccggaattccgaattaatacgactcactatagggctatcagccaaactgtgc

GFP-RNAi 5'- gaattaatacgactcactatagggtgagcaagggcgaggagctg

## Methods

### Conditional expression of transgenes in specific gut cells

Conditional expression of indicated transgenic lines in specific cells was obtained by activating a UAS-transgene under the expression of the indicated cell-specific Gal4-driver together with the tub-Gal80$^{ts}$ construct (*Jiang et al., 2009*). Flies were raised at 18°C. At 2–4 days, F1 adults female progeny (and males in experiment depicted in *Figure 1—figure supplement 2A, B*), were transferred to the restrictive temperature 29°C (Gal80 off, Gal4 on) for two days unless indicated otherwise. Next, guts were dissected and analyzed as described below. At least three biological independent repeats

were performed for each experiment. Where possible, we used multiple RNAi lines, as indicated below and in the figure legends. Specifically, in the case of Hey targeting, we used the following lines in specific experiments: *Figure 1*: G-H' VDRC#103570, termed Heyi, I, I' NIG-FLY: 11194 R-1. Identical phenotype was also observed using NIG-FLY: 11194 R-3, and a line combining the Bloomington lines #31898 and #41650 (not shown). Figure S1 A-C' Heyi, D' NIG-FLY: 11194 R-1. *Figure 2*: Heyi, C, C', E, E'; 11194 R-1, D. *Figure 3*: 4: Heyi. and NIG-FLY:.

## Regional-Gal4 targeting in ECs

The original description of this line is available at http://flygut.epfl.ch/patterns/1086. It is derived from the regulatory region of CG9003 that we did not identified as a Hey target. In addition to the published data, we further characterized the expression of this regional line and validated that it is only expressed in mature ECs, and that control targeted region is identical to non-targeted region (Figs. S7A-D). Quantification of this set of experiments was performed by comparing phenotypes of the targeted region to targeted region expressing LacZ control (Fig. S7Q).

*Conditional G-TRACE analysis* was performed using Myo-Gal4; G-TRACE flies were crossed to UAS-LacZ; Gal80$^{ts}$ (control) or UAS-Hey-RNAi; Gal80$^{ts}$ and the appropriate genotypes were raised at 18°C (a temperature at which no G-TRACE signal was detected). At 2–4 days, adult females were transferred to 29°C and linage tracing was performed.

*Paraquat treatment:* Where indicated, adult females were exposed for 48 hr to 5% sucrose solution with, our without, 5 mM paraquat similar to the treatment performed by *Chatterjee and Ip (2009)*.

## Gut dissection and immunofluorescence detection

Gut fixation and staining were carried out as previously described (*Shaw et al., 2010*).

## Clonal analysis of Hey-mutant clones

The MARCM technique was used to study positively marked (GFP) *Hey$^P$* mutant clones as described by *Bardin et al. (2010)*.

## Quantification by IMARIS and 3d reconstruction

Z-stacks from Zeiss LSM 700 images were reconstructed into 3D images using IMARIS software (Version 8.3.0, Bitplane, Switzerland). Image analysis using a surface modulus algorithm was performed to measure the penetration depth of the cells. A single and representative midgut was used to reconstruct the 3D image of the cells. Blending calculations enabled to adjust the transparency of each channel (one channel per fluorophore), making it possible to view the composition of the structures. After generating a surface object, the same software automatically calculated a range of statistical parameters including surface area, volume, and DAPI and antibody intensity of the different midgut cells.

## Gut integrity and animal survival

Two to four days old female flies from the specified genotype were collected into a fresh vial (10 flies per vial). All flies were kept in vials in a humidified, temperature-controlled incubator at 29°C. Flies were transferred into vials containing fresh food every two days and were scored for viability. Statistical analyses and overall survival curves were analyzed using GraphPad Prism software.

## 5mC-Chromatin Accessibility

Females guts were dissected in Schneider medium, washed in PBS, and fixed in 6% EM-grade formaldehyde (Polysciences, Warrington, PA) diluted in PBS, with three volumes of heptane for 20 min. The dissected tissue was then washed with PBS, permeabilized using 0.5% Triton X-100 in PBS for 1 hr, washed with PBS three more times and blocked in PAT (1% BSA, 0.1% Triton X-100 in PBS (PBX) overnight. The next day, guts were washed twice with 250 µl M.SssI reaction buffer supplemented with 16 µM S-Adenosyl-L-methionine, Rre-suspended in 50 µl of M.SssI reaction buffer supplemented with 25U of M.SssI (NEB, Ipswich,MA) and 16 µM S-Adenosyl-L-methionine and incubated for 1 hr at 25°C on an orbital shaker. DNA was denatured by adding 1 ml of 2N HCl for 30 min at room temperature. The solution was then neutralized in 100 mM Borax for 5 min and washed twice

with PBS. Detection of 5mC was performed using anti methyl-cytosine antibody primary antibody (monoclonal anti-5-mC, clone 33D3, Active Motif, Carlsbad, CA) that was added at 1 µg/ml. The next day, guts were washed three times with PBT (0.1% BSA, 0.1% Triton X-100 in PBS). We used secondary anti-mouse used Alexa-568 (1:1000) and DNA was visualized using DAPI for 1 hr and following two washes was mounted in Fluoromount-G for confocal imaging.

## DamID, expression signatures and bioinformatics analyses

DamID was performed as described below, following *Rincon-Arano et al. (2012)* and *Singer et al. (2014)*.

## DamID chromatin profiling

Dam-Hey was generated by cloning Hey cDNA pNDamMyc vector to generate Dam-Hey (FL) and verified by western blot and immunostaining. DamID chromatin profiling was performed as previously described (*Bianchi-Frias et al., 2004*; *Singer et al., 2014*). Dam-methylated DNA was hybridized on NimbleGen DM2 CGH arrays with a probe spacing of ~300 bp (*Filion et al., 2010*). DamID profiles for each Dam fusion protein were performed in duplicates, plus an additional dye swap technical replicate. Dam fusions were compared with methylase alone to control for non-specific accessibility. Microarray data was processed as previously described (*Guelen et al., 2008*; *van Bemmel et al., 2010*; *van Bemmel et al., 2013*). All data processing and analyses were performed using the R package for statistical computing (http://www.r-project.org). Data was LOESS normalized and a custom R script was implemented to define overlapping domains using a minimum 80% overlap threshold according to which at least one domain had to have at least 80% overlap with the other. We used parameter optimization functions within CGTools to determine the transition threshold and proportion of positive probe thresholds. Sharp transitions in the DamID signal were identified using a sliding edge filter (window size 199 probes) and adjacent transitions exceeding a threshold (here 0.3) were combined into domains if at least 70% of the enclosed probes had a positive log2 ratio.

## Purification and isolation of gut cells

To isolate EBs we adopted a surface marking followed by magnetic cell purification protocol used to isolate border cells (*Wang et al., 2006*). Specifically, *Hey-Gal4* >UAS mCD8GFP virgin female flies were crossed (at 18 °C) to either UAS-GFP or UAS–Hey RNAi males. Two to four-day old female progeny were collected and aged at 18 °C for 2–10 days and then transferred to 29 °C for 2 more days prior to dissection. Purification protocol was carried out as previously described (*Wang et al., 2006*). Enrichment was determined in each experiment using FACS and targeting was confirmed by immunostaining.

## RNA extraction, cDNA preparation, and gene expression analysis

RNA was isolated in triplicate from independent fly crosses for each experiment and was prepared from purified cells according to the Qiagen RNeasy protocol. A total of 50 ng of total RNA sample was used as starting material to make cRNA probes followed by NimbleGen probe labeling. Double-stranded cDNA synthesis was carried out using Superscript III (Invitrogen 18080–051) followed by cRNA amplification using an Epicenter TargetAmp 1-Round aRNA amplification kit (TAU1R5124). Double-stranded cDNA was synthesized again and further labeled using a NimbleGen One-Color DNA labeling kit (05-223-555) and then hybridized to a *D. melanogaster* Gene Expression 4 × 72K Array (A4509001-00-01) according to the NimbleGen expression array protocol. Arrays were scanned using a GenePix 4000 microarray scanner and data was extracted using NimbleScan v2.4 software (Roche NimbleGen Inc, Madison, WI, USA). Microarray data was processed in NimbleScan using default settings. Data was further uploaded into the ArrayStar software version 3 (DNASTAR, Inc, Madison, WI, USA) for normalization and statistical analysis. A moderated t-test was performed with false discovery rate (FDR; Benjamini–Hochberg), and multiple testing corrections were applied on data ($p<0.05$) with a 2-fold change or greater.

## Bioinformatics analyses

LiftOver to dm3 was applied to all dm2 genomic data. End analyses (meta-analysis) were performed as described by *Deal et al. (2010)*, using custom scripts in R or Galaxy/Cistrome (*Shin et al., 2009*;

*Liu et al., 2011*). Visualization was carried out using the UCSC browser (*Karolchik et al., 2014*). For Hey enrichment on states defined by *Filion et al. (2010)* and modENCODE (2010), Hey mean signal was averaged along each state/region and plotted as boxplot in R. Six thousand random sequences ranging between 2–5 Kb were generated in XLSTAT, considering a similar number of binding sites for regulatory elements revealed by *Filion et al. (2010)* and *Kharchenko et al. (2011)*. Meta-analyses for histone modifications were performed by aligning all regions at their center and averaging the normalized mean probe values as a function of distance. Meta-analyses were performed on individual experiments, the signal was averaged, and the standard errors were calculated in Excel. Similar analyses were performed on all *Drosophila* Ref-seq genes at both their transcriptional start and termination sites (*Deal et al., 2010*). Additional data source: Chromatin states defined by the mod-ENCODE Consortium as well as Kc167 RNA-seq data were obtained from: www.modencode.org/publications/integrative_fly_2010, (*Gerstein et al., 2010*); GSE22069 (*Filion et al., 2010*).

## RNA-seq analysis of LamDm0 expression in ECs

Four independent biological repeats were performed. Quality measurements for total RNA were performed using the TapeStation 2200 (Agilent). Library prep and data generation of RNA-sequencing: Starting with 100 ng total RNA, twelve RNA-seq libraries were produced using the NEBNext Ultra Directional RNA Library Prep Kit for Illumina (NEB, cat no. E7420), according to manufacturer's protocol. mRNA pull-up was performed using the Magnetic Isolation Module (NEB, Cat No. E7490). Two of the twelve libraries (Samples B1 and B2) were disqualified due to low library yield and high levels of adaptor dimer. Equal molar concentrations of the remaining ten libraries were mixed in a single test tube. The RNA-seq data was generated on two lanes of HiSeq2500, 50 SR. NGS QC, alignment and counting 50 bp single-end reads were aligned to *Drosophila* reference genome and annotation file (*Drosophila melanogaster.* BDGP6 downloaded from ENSEMBL) using TopHat (v2.0.13), allowing for two mismatches per read using the 'very-sensitive' option. The number of reads per gene was counted using Htseq (0.6.0).

For LamDm0 RNA-seq experiment and to generate descriptive and DEGs analysis: Sample clustering and differential expressed genes (DEGs) were calculated using Deseq2 package (version 1.10.1).

Gene ontology analysis of Hey DamID overlapping genes and mRNA expression was performed using the Database for Annotation, Visualization and Integrated Discovery (DAVID http://david.abcc.ncifcrf.gov/home.jsp; v6.7 and v6.8) using default settings (2-fold p<0.05 Benjamini P value for analysis E-03) (*Huang et al., 2009*).

Biological processes with a p-value lower than 0.05 were further analyzed with Revigo (*Supek et al., 2011*). Gene ontology analyses via Cytoscape: ClueGO app (v2.2.5) in Cytoscape (v 3.4.0) was used to conduct GO enrichment analyses. In our study, ClueGO was used to identify different functional groups in the following terms: Biological Process (BP), Cellular Component (CC) and Molecular Function (MF) enrichment analysis. A p-value≤0.05 was used as the cut-off criterion. In addition expression of transcriptional analyses were compared to gut specific signature obtained from flygut (https://flygut.epfl.ch/expressions).

*RNAi in S2 cells:* RNAi in S2 cells was performed as described by *Orian et al. (2007)*.

## Acknowledgements

We are grateful to Susan Parkhurst, Alan Schwartz, Yossi Gruenbaum, Ze'ev Paroush, and Peleg Hasson for their critical comments and suggestions, and to Rotem Meller for the graphics. We would like to thank Sarah Bray, Ali Shilatifard, Adi Salzberg, Bruce Edgar, Gregory Hannon, Tao-shih Hsieh, Jeff Reinitz, Lori Wallrath, Pamela Geyer, Yossi Gruenbaum, Lorry Pile, Erika Matunis, Rongwen Xi, and the Bloomington, VDRC, and NIG-FLY *Drosophila* stock centers for sharing antibodies, fly lines, reagents, and data. We are thankful to Bas van Steensel for reagents and for sharing microarray designs and scripts, and to Dave Scalzod and Ryan Basom of the Fred Hutchinson Cancer Research Center for their help with the bioinformatics analyses.

This research was supported by the German Research Foundation (SBF 688, TP A16) (to MG), the Israel Science Foundation (ISF) (Grant 739/15), and the Rappaport Institute for Research in the Medical Sciences and Flinkman-Maraendy cancer research grant (to AO).

## Additional information

### Funding

| Funder | Grant reference number | Author |
|---|---|---|
| Israel Academy of Sciences and Humanities | 739/15 | Amir Orian |
| Deutsche Forschungsgemeinschaft | SBF 688 | Manfered Gessler |
| Flinkman-Marandy Cancer Research Grant | 0001 | Amir Orian |
| Deutsche Forschungsgemeinschaft | TPA 16 | Manfered Gessler |

The funders had no role in study design, data collection and interpretation, or the decision to submit the work for publication.

### Author contributions

Naama Flint Brodsly, Data curation, Formal analysis, Investigation, Methodology, Writing—original draft, Writing—review and editing; Eliya Bitman-Lotan, Data curation, Formal analysis, Investigation, Methodology, Writing—original draft, Project administration, Writing—review and editing; Olga Boico, Adi Shafat, Data curation, Formal analysis, Investigation; Maria Monastirioti, Resources, Formal analysis, Investigation, Methodology; Manfred Gessler, Conceptualization, Writing—original draft, Writing—review and editing; Christos Delidakis, Conceptualization, Investigation, Writing—original draft, Writing—review and editing; Hector Rincon-Arano, Resources, Data curation, Formal analysis, Investigation, Methodology, Writing—original draft; Amir Orian, Conceptualization, Formal analysis, Supervision, Funding acquisition, Investigation, Writing—original draft, Writing—review and editing

### Author ORCIDs

Manfred Gessler https://orcid.org/0000-0002-7915-6045
Amir Orian https://orcid.org/0000-0002-8521-1661

### Decision letter and Author response

Decision letter https://doi.org/10.7554/eLife.44745.050
Author response https://doi.org/10.7554/eLife.44745.051

## Additional files

### Supplementary files

• Source code 1. Script code for comparing DamID binding to genomic data.
DOI: https://doi.org/10.7554/eLife.44745.035

• Supplementary file 1. Gene expression analyses 1. Progenitor cells differentiation to ECs 2. Hey-regulated genes in whole guts 3. Hey-regulated genes in purified progenitors. 4. RNA-seq of genes regulated by LamDm0 expression in ECs
DOI: https://doi.org/10.7554/eLife.44745.036

• Supplementary file 2. Go analysis of DEG in Hey RNAi and LaminDm0 O .E (by DAVID and Cytoscape). GO analyses GO analysis of Hey RNAi in whole gut and purified progenitors by DAVID GO analysis of Hey RNAi in whole gut and purified progenitors by cytoscape GO analysis of genes regulated by LamDm0 expression in ECs
DOI: https://doi.org/10.7554/eLife.44745.037

• Supplementary file 3. Genomic regions bound by Hey-DamID.
DOI: https://doi.org/10.7554/eLife.44745.038

• Supplementary file 4. Shared putative-Hey direct targets in progenitors and ECs.
DOI: https://doi.org/10.7554/eLife.44745.039

• Supplementary file 5. GO analysis for shared regulated DEGs between Hey-RNAi to ectopic expression of Esg in ECs.
DOI: https://doi.org/10.7554/eLife.44745.040

• Supplementary file 6. Hey-regulated putative digestive enzymes.
DOI: https://doi.org/10.7554/eLife.44745.041

• Transparent reporting form
DOI: https://doi.org/10.7554/eLife.44745.042

## Data availability

Gene expression analysis was deposited at GEO (GSE87896), LaminDm0- RNA-seq experiment was deposited at GEO (GSE112640) and GSE Custom R script mentioned in Methods subsection DamID chromatin profiling is available as Source code 1.

The following datasets were generated:

| Author(s) | Year | Dataset title | Dataset URL | Database and Identifier |
|---|---|---|---|---|
| Orian A, Olga Boico, Hector Rincon-Arano Bitman-Lotan E | 2016 | Expression profiling by array and Genome binding/occupancy profiling by genome tiling array | https://www.ncbi.nlm.nih.gov/geo/query/acc.cgi?acc=GSE87896 | NCBI Gene Expression Omnibus, GSE87896 |
| Orian A, Flint-Brodsly N, Bitman-Lotan E | 2018 | RNAseq analysis of whole Guts over expressing LaminDm0 or GFP in Enterocytes | https://www.ncbi.nlm.nih.gov/geo/query/acc.cgi?acc=GSE112640 | NCBI Gene Expression Omnibus, GSE112640 |

The following previously published dataset was used:

| Author(s) | Year | Dataset title | Dataset URL | Database and Identifier |
|---|---|---|---|---|
| Filion GJ, van Steensel B et al | 2010 | Protein profiling reveals five principal chromatin types in Drosophila cells | https://www.ncbi.nlm.nih.gov/geo/query/acc.cgi?acc=GSE22069 | NCBI Gene Expression Omnibus, GSE22069 |

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
