## [Decision Letter]

Thank you for submitting your work entitled "The transcription factor Hey and nuclear lamins specify and maintain cell identity" for consideration by *eLife*. Your article has been reviewed by two peer reviewers, and the evaluation has been overseen by a Reviewing Editor and a Senior Editor. The reviewers have opted to remain anonymous.

Our decision has been reached after consultation between the reviewers. Based on these discussions and the individual reviews below, we regret to inform you that your work will not be considered further for publication in *eLife*. If you are able to address all the main issues raised by the reviewers we may consider a new version as a new submission.

*Reviewer #1:*

The authors investigate the role of Hey, a bHLH transcription factor that is related to the Notch pathway, in differentiation of Enterocytes (EC) cells in the adult *Drosophila* midgut. Hey appears to play a Notch independent role, in maintaining the differentiation state of ECs. Following a combination of genetics, cell biological and bioinformatics methods, the authors conclude that Hey is crucial in mediating and maintaining EC differentiation (in addition to its context specific role in other cell types of the lineage, i.e. EB cells). The story culminates to the point where Hey regulates a developmental switch in lamins, lamDm0 and lamC, which are expressed in EBs and terminally differentiated ECs respectively. In fact the authors manage to show that:

a) Lamins appear to share a network of target genes they coregulate with Hey, albeit they do so via a different mechanisms, i.e. by maintaining nuclear architecture as opposed to Hey maintaining Acetylation of histones and

b) RNAi knockdown or overexpression of lamDm or lamC confirms the directionality of action in the lamins-Hey circuitry.

The manuscript is full of data, but the organization of the presentation was a bit difficult to follow. Nevertheless, the data are clear and they create new possibilities for future experiments. Indeed, the authors have managed to use existing tools and even make new ones (Hey MARCM stock, Hey Gal4, etc.) that can benefit their future studies as well as the studies of other groups looking into stem cell biology or maintanance of differentiation.

Major point:

Can the authors rescue Hey-RNAi phenotypes, such as the reduced survival and leakage of food in the abdominal cavity, by repressing laminDm or overexpressing lamin C?

Does overexpression of lamDm0 or repression of lamC cause phenotypes like Hey RNAi with regard to gut homeostasis, such as STAT reporters activity, Notch activity, upregulation of Dlg and Arm?

*Reviewer #2:*

This study by Orian and colleagues examines phenotypic consequences of hey perturbation in the adult *Drosophila* midgut and transcriptional control by Hey in the midgut and in cell culture. The authors chief conclusions are that (1) Hey is a core terminal transcription factor for maintaining enterocyte identity and that (2) Hey executes this function by remodeling the nuclear architecture of enterocytes via control of lamin switching.

Overall, I feel that the data presented--although substantial in volume--unfortunately fall far short of providing convincing support for these conclusions. In my view, the revisions necessary to substantiate the authors' claims would be quite a major undertaking.

My concerns involve what I see as significant issues regarding data interpretation and study design:

1) Hey acts broadly in multiple cell types, but the study does not offer insight into how Hey achieves cell type-specific regulation. Specifically, Hey is expressed in all midgut cells and has pleiotropic roles in stem cells, enteroblasts, and enterocytes. Without knowing the mechanisms that underlie cell type specificity, I am not sure whether Hey is actually a core transcription factor or merely an accessory co-factor.

2) A major element of the authors' model is that the switch from LamDm0 to LamC during enterocyte differentiation produces widespread changes in gene expression by promoting different nuclear architectures. However, the study does not include any assessment of nuclear organization, lamina-chromatin association, or other lamin-influenced structural feature for any midgut cell type (either normal or genetically manipulated enterocytes and enteroblasts). As far as I can tell, the authors' conclusions that Hey and LamC "remodel the nuclear architecture" and "determine a nuclear organization that is unique to ECs", although provocative, are not supported by either direct or indirect evidence.

3) The datasets that are used to draw conclusions about cell type-specific transcriptional signatures and enhancer activity are not themselves specific to the relevant cell types.

a) To isolate enteroblasts for microarray analysis (Figure 4A), the authors sort cells for UAS-GFP under control of a GAL4 driver comprising the 3'UTR of hey. They state that this driver is expressed "predominantly" in enteroblasts. However, many Δ-expressing stem cells appear to express hey>GFP (Figure S4A), and many cells that express hey>GFP do not express the enteroblast-specific Notch reporter (Figure S4B). These features raise questions as to the specificity of the "enteroblast" datasets.

b) Hey enhancer activity is inferred from DamID of cultured Kc167 cells, not of midgut cells. The authors state they were unable to perform DamID in midgut cells (although some precedents do exist: Korzelius et al., 2014, Loza-Coll 2014, Jin 2015, Chen 2018). Given that Hey has different targets in different cell types even within the midgut, it seems tenuous to extrapolate enhancer binding in Kc167 cells to midgut enterocytes.

4) It is not clear to me that hey-knockdown enterocytes truly "lose their differentiated identity", as stated by the authors and as would be expected if Hey is a core terminal factor.

a) Morphologically, the knockdown enterocytes still resemble enterocytes and not any other cell type.

b) Physiologically, not evidence that enterocytes downregulate digestive enzymes or nutrient transporters or otherwise attenuate their digestive function.

c) The phenotypic effects of hey knockdown (e.g. elevated JAK-STAT, expression of Notch reporters, loss of H3K27Ac, lamin mis-expression) appear partial, which is inconsistent with the notion that Hey is centrally important. (These partial effects do not appear to correlate with the extent of Myo1A driver activity, as far as I can gauge by co-expression of GFP in the images that are shown.)

d) Many effects of hey knockdown resemble the well-characterized hallmarks of regeneration and ageing (e.g. elevated JAK-STAT, expression of Notch reporters, disorganization, leakiness). Indeed, I might interpret the hey knockdown phenotype as simply recapitulating the normal enterocyte response to damage and stress-not as wholesale reprogramming of enterocyte identity.

[Editors’ note: what now follows is the decision letter after the authors submitted for further consideration.]

Thank you for submitting your article "The transcription factor Hey and nuclear lamins specify and maintain cell identity" for consideration by *eLife*. Your article has been reviewed by two peer reviewers, and the evaluation has been overseen by a Reviewing Editor and K VijayRaghavan as the Senior Editor. The reviewers have opted to remain anonymous.

The manuscript went back for review to the previous reviewers but one reviewer was unable to re-review the manuscript. We therefore decided to send it out to another reviewer. Based on the comments of this reviewer we believe that upon making the suggested changes the manuscript may be suitable for publication. However, we would like to see a detailed rebuttal. The new reviewer provides many useful suggestions, particularly on data quantification and on the writing, that should be addressed.

*Reviewer #2:*

The manuscript of Flint-Brodsly et al. is an exciting manuscript. The authors have improved the flow of the manuscript and strengthened their conclusions, by performing multiple additional experiments, while correcting the phrasing of certain points throughout the manuscript at the same time.

The manuscript describes the pleiotropic role of Hey TF in the *Drosophila* gut, focusing mostly on its role in the active maintenance of the differentiated state of the EC cells. The techniques the authors used range from genetic and cell biology experiments to multiple RNAseq and DamID experiments. The authors have addressed this reviewers' remarks.

Additional data files and statistical comments:

A major problem that I have in evaluating this work is the general lack of quantification throughout the manuscript. The adult gut is a very dynamic tissue that responses rapidly to subtle environmental changes. Because of this, it is especially important that careful quantification be performed as it is very rare that any phenotype is black and white. Therefore, while there are some potentially interesting findings (changes in Lamins during differentiation; a role of Hey in EC LamC expression; potential regulatory roles for Lamins in controlling fate genes), it is impossible to judge from the data presented whether the anecdotal images are representative of all guts, all regions of the gut, males or females, etc. Ideally, all data should be rigorously quantified, with data points plotted--not just an average, and compared with appropriate statistics. In particular, the effect on Pdm1 expression of Hey knockdown with Myo1AGal4 looks more subtle than some other phenotypes, which raises the issue of how direct or indirectly certain phenotypes are. Proper quantification of the data will make this more clear.

Another major issue that I have is that because the RNAi of Hey in ECs generates a proliferative response, many of the effects the authors detect could be secondary due to this. This should be addressed.

Finally, the paper could be more streamlined to improve readability. There are some things that seem irrelevant and could be removed (Figure 2E', Figure 6S, T). Also, it seems to me that the characterization of the proliferative response in Figure 6 needs to come much earlier in the manuscript as it is important to interpret the rest. The "hyperplasia" is mentioned but not shown then or quantified.

1) Quantification throughout, n= should be somewhere in legends or figures.

*–* Figure 1B-D. What% of cell types are marked by Hey?

*–* Figure 1E, E', G, G': How frequent is the large Dl+ cell phenotype. One example is shown- please quantify% of Dl+ cells that are Pdm1+. This phenotype might be linked to the proliferative state of the gut producing more newly made ECs. Their data do not show that the EC which downregulates Hey actually re-expresses Dl. Another hypothesis is that Hey knockdown in ECs triggers stress and activates ISC proliferation, in which conditions ISCs are known to be enlarged (for example upon yki overexpression, see Shaw et al., 2010, Figure 1H-J).

*–* Figure 1F, F', H, H4- what% of ECs lose Pdm1 staining up Hey RNAi expression compared to controls? Can they co-stain with Hey to see if there is a direct relationship between Hey disappearance and Pdm1 disappearance?

*–* Figure 2 F, Figure 5I: the n should be put in the figure.

*–* Figure 3 D-G: What% of Dl+ cells express RecQ4 in Myo1A>Hey RNAi and please compare to controls.

*–* Figure 4C, C'; Figure 5A-B- Importantly a control is missing: the same region of guts expressing no RNAi. Due to the regional differences, I am not convinced one can compare across compartment boundaries like in the wing disc. Please add control and quantify% of cells altered for both.

*–* Figure 4D-G: These phenotypes could be due to differences in proliferation between HeyRNAi and wild-type. Newly produced ECs will inherit LacZ and may have more accessible chromatin. What does the LacZ line look like in wild-type MARCM clones that would similarly be proliferative tissue? In any event, these phenotypes need quantification.

*–* Figure 4I-CC; Figure 5G-Q: Please show the% of cells show these phenotypes, compared to controls, with stats.

*–* Figure 6: some quantification must be done here. The effect on MESH looks much stronger than the effect of Hey knockdown, in terms of% of ECs losing Hey. Why is this? Quantification of PH3 with >15 guts.

*–* Figure 6UV: They state in the legend that 22% of flies have a smurf phenotype upon Hey RNAi. It would be helpful to write this on the figure so it is clear.

*–* Idem for the rest of the figures and supplementary figures.

2) An important part of the manuscript is the finding that Lamin types switch from the progenitors to the EC and that Hey is required for maintenance of expression of the LamC in the EC. This finding seems to directly contradict recent data of Petrovsky and Großhans, 2018 that observe that LamC is uniformly expressed in the epithelial cells of the *Drosophila* intestine. What is the difference in experimental set up or interpretation? The authors should minimally discuss this contradictory finding that goes against their model.

3) Important interpretations were made using GO term analysis. I am concerned that the results obtained are biased toward genes expressed in the ECs. My understanding is that when performing GO term analysis, the background gene set should be the expressed genes in the tissue or cell type. If this is not used, a bias will be obtained towards these genes since, the genes altered are those expressed in the tissue. The authors should use the background gene set from the gut epithelia plus surrounding muscle, which can be obtained at Flygut-seq for the analysis in Figure 3 and S8 to see if the categories of genes altered in HeyRNAi or LamDm0 overexpression are still statically enriched over the gene set of intestine genes (and not the whole genome).

4) I feel that the authors overinterpret the data presented in Figure 2A-I'. It is also written in a very confusing manner- please start by outlining the experiment and the expected results, then give the results without an interpretation of what the different colors of cells are. Then speculate on the potential meaning. They should not refer to the GFP+RFP- cells as "no longer fully differentiated". Similarly, do not refer to the RFP+GFP- cells as "early ECs" or the GFP-RFP- cells as "potentially mis-differentiated". These interpretations should be kept separate from the data and then substantiated with additional markers. A timecourse would be helpful as these populations should evolve over time. Also, given the difference in proliferation of control and knockdown, could the results be explained by altered kinetics in a regenerating gut? They could see if infection or damage conditions show a similar effect, which would argue against their model.

-Similarly, I believe they have misinterpreted the phenotype of Hey RNAi with respect to Notch signaling. Discussion, "The ectopic activity of a Notch-reporter observed in EC-NFD upon Hey targeting in ECs suggests that Hey limits Notch activity in ECs." This is likely only due to a secondary effect on causing stress in the gut, driving ISC proliferation, and therefore making more, new EB cells that express the Su(H)-GBE reporter. It is unclear in the gut whether Hey has any impact on Notch signaling, is regulated by Notch, or is simply parallel.

5) Did the authors show data of LaminC RNA decrease in S2 cells?

6) "These phenotypes were suppressed by over-expression of UAS-Hey, but not by a control transgene (Figures S1C, C4). Moreover, RNAi knockdown of Notch, Suppressor of Hairless, other HES TFs, and HES-related cofactors did not phenocopy Hey loss in ECs (Figures S2A-J)."

-what phenotypes are being assessed. Please show some quantification.

7) It should be clearly stated in the text that microarrays are used to assess transcriptomes. Quantification of the numbers of hey-Gal4 expressing cells should be done in the different backgrounds (controls vs heyRNAi). PCA plots should be shown.

8) P7 "Overall, the transcriptional signature of guts in which Hey was targeted in ECs largely resembled the signature of control purified progenitors"

- "Hey knockdown was induced" would be more clear

Also, it is not at all clear to me what is being compared here: is it the genes that change in Myo1A>GFP vs Myo1A>hey RNAi compared to the genes that are expressed in "heyGal4+ cells"? What are the heyGal4+ cells compared to? Whole guts? These are primarily EBs. The interpretation of these data are not clear to me? The microarray of whole gut while knocking down in ECs will give ISC, EB and EE genes altered in a non-autonomous fashion as well as EC genes altered.

9) As mentioned above, quantification of the effects seen in Figure 4F-G; What% of guts show the effect? Figure 4 I-CC, what% of ECs show these effects? There is undoubtedly variability in these data- what is the level? Could these effects be due to different ages of EC cells? More new EC cells are likely made in Hey knockdown conditions- is it only new EC cells that show this? They could for example make MARCM clones and look at 3 day old ECS in wild-type to see if the chromatin marks are established or if they come during maturation of the EC.

10) Figure 5G-I- they should also quantify what happens to Dl+ cells- here they lose both Dl+ cells and LaminDm0+ esg+ cells.

11) The data in 5J are not convincing on their own. This does not look like Dl staining, which is usually vesicular. Please show some additional images and quantification. Similarly, 5M is not convincing: in this context, clearly there are more ISC divisions, leading to more esg+ cells, which are likely not all ISC/EBS, but newly produced ECs, which therefore express Pdm1. How do the authors identify progenitors in Figure 5M?

12) PCA plots of the RNAseq data should be shown. They should state to what they are comparing the Myo1A>LamD0 data (UAS-GFP?).

13) I disagree with the following statement in the Introduction and Discussion, that I do not feel is supported by their references:

"Indeed, multiple lines of evidence have established that failure to maintain a differentiated identity is a hallmark of aging…" (Schwitalla et al., 2013; Deneris and Hobert, 2014; Ocampo et al., 2016)

To me, none of these references really strongly support the idea that differentiated cell fates decline during normal aging, which is what stated as fact. I would not say that this is a well-established notion. However, one paper that does support this is from the fly gut field and is not cited (Li, Qi and Jasper, 2016).

-In the Discussion section on Cell identity, aging, and pathological plasticity. "During *Drosophila* and vertebrate aging, or under various stress condition, differentiated cells in the gut, pancreas, and trachael epithelium behave similarly, reflecting a general principle." While, I agree published data suggest plasticity in cancer or pathological cell loss in these tissues, a function of loss of identity during normal aging is less obvious to me and should be better supported by published literature. Mutant contexts such as Progeria syndromes are not equivalent to wild-type aging.

[Editors' note: further revisions were requested prior to acceptance, as described below.]

Thank you for resubmitting your work entitled "The transcription factor Hey and nuclear lamins specify and maintain cell identity" for further consideration at *eLife*. Your revised article has been favorably evaluated by K VijayRaghavan (Senior Editor), a Reviewing Editor, and two reviewers.

The manuscript has been improved but there are some remaining issues that need to be addressed before acceptance, as outlined below. The proposed experiment at the end of this reviewers critique seems easy to do and should be informative.

*Reviewer #3:*

The revised manuscript by Flint Brodsly and colleagues has been greatly improved. A major effort has been made to add much of the needed quantification.

However, I still feel that all of the effects described could be due to increased stem cell proliferation and rapid production of EC cells. The authors did not address this point of mine, instead of saying numerous times that this was outside of the scope of the manuscript.

For example:

"Another major issue that I have is that because the RNAi of Hey in ECs generates a proliferative response, many of the effects the authors detect could be secondary due to this. This should be addressed.”

*We fully agree. This is now better addressed including our G-TRACE analysis, as well as in the discussion. The potential cross-talk from ECs that lack Hey or LamC to progenitors or during aging is a broad topic that we are studying. I feel that the experimental molecular details; such as the identity of ligands secreted from ECs and that impinge and induce on progenitor regeneration is outside of the scope of this current manuscript (that is already dense), as these experiments will require at least six months (obtaining stocks and the subsequent genetics required).*

Also, given the difference in proliferation of control and knockdown, could the results be explained by altered kinetics in a regenerating gut? They could see if infection or damage conditions show a similar effect, which would argue against their model.

*Our manuscript focused on how ECs maintain identity, but rather in young adults or during normal aging. We strongly feel that how cell identity is maintained upon infection or DNA damage – while highly interesting – is outside of the scope of the present study.*

- Similarly, I believe they have misinterpreted the phenotype of Hey RNAi with respect to Notch signaling. Discussion, "The ectopic activity of a Notch-reporter observed in EC-NFD upon Hey targeting in ECs suggests that Hey limits Notch activity in ECs." This is likely only due to a secondary effect on causing stress in the gut, driving ISC proliferation, and therefore making more, new EB cells that express the Su(H)-GBE reporter. It is unclear in the gut whether Hey has any impact on Notch signaling, is regulated by Notch, or is simply parallel.

*The reviewer makes a good point. We have changed the text to better reflect this point:*

*"The observed Notch-reporter activity in PPCs upon Hey targeting in ECs, may reflect stress-increased ISC proliferation and Notch activation in enlarged progenitors. Alternatively, this ectopic activity may suggest that Hey limits Notch activity in ECs."*

What I was suggesting that the authors do was not a six month experiment, but a simple treatment of flies with some damaging agent that induces a similar proliferative response to that seen upon Hey RNAi in ECs (one of: Ecc15, Pe, paraquat), and assess some of their nuclear phenotypes (LamC, LamDm0). If they do not see an impact on these marks, this would definitively rule out that the effect of Hey knockdown on ECs is merely indirect through rapid production of new ECs. For example, the effect on Arm that they find is also seen upon increased proliferation in the gut. Newly-made ECs upon rapid ISC proliferation are different from those made under homeostatic conditions. I do realize that this is extra work, but honestly, if this were my own paper, I would at least want to know this result and comment on whether or not this is likely indirect via proliferation stimulation. The alternative which may be more complicated genetically is to block proliferation in the Hey RNAi contexts and see if the effects on LamC or LamDm0 are still seen.

---

## [Author Response]

Reviewer #1:

The authors investigate the role of Hey, a bHLH transcription factor that is related to the Notch pathway, in differentiation of Enterocytes (EC) cells in the adult Drosophila midgut. Hey appears to play a Notch independent role, in maintaining the differentiation state of ECs. Following a combination of genetics, cell biological and bioinformatics methods, the authors conclude that Hey is crucial in mediating and maintaining EC differentiation (in addition to its context specific role in other cell types of the lineage, i.e. EB cells). The story culminates to the point where Hey regulates a developmental switch in lamins, lamDm0 and lamC, which are expressed in EBs and terminally differentiated ECs respectively. In fact the authors manage to show that:a) Lamins appear to share a network of target genes they coregulate with Hey, albeit they do so via a different mechanisms, i.e. by maintaining nuclear architecture as opposed to Hey maintaining Acetylation of histones andb) RNAi knockdown or overexpression of lamDm or lamC confirms the directionality of action in the lamins-Hey circuitry.The manuscript is full of data, but the organization of the presentation was a bit difficult to follow. Nevertheless, the data are clear and they create new possibilities for future experiments. Indeed, the authors have managed to use existing tools and even make new ones (Hey MARCM stock, Hey Gal4, etc.) that can benefit their future studies as well as the studies of other groups looking into stem cell biology or maintenance of differentiation.

We have extensively revised and re-wrote entire manuscript, simplified the presentation, reduced data load, revised supplemental figures and data. We were very strict on data interpretation. We have removed parts of the genomic data, that while interesting, distracted the reading flow.

Major point:Can the authors rescue Hey-RNAi phenotypes, such as the reduced survival and leakage of food in the abdominal cavity, by repressing laminDm or overexpressing lamin C?Does overexpression of lamDm0 or repression of lamC cause phenotypes like Hey RNAi with regard to gut homeostasis, such as STAT reporters activity, Notch activity, upregulation of Dlg and Arm?

We have performed the suggested experiments:

1) As expected, we demonstrate that perturbation in Lamin expression resulted in changes in gut homeostasis including ectopic activation of signaling pathway, as well as distinct changes in expression of cell adhesion molecules, loss of gut integrity and overall survival (New Figure 6). We specifically show that loss of LamC results in ectopic JAK-STAT activity, as well as expression of the stem cell related adhesion co-factor Armadillo (Arm), but had no effect on the expression of EC adhesion molecules like MESH. Inversely expression of LamDm0 in ECs resulted in reduced expression of EC genes like MESH but had no effect on Arm expression. We also show that expression of LamDm0 in EC, reduces the overall lifespan. Moreover we show that this regulation of lamins expression is impaired during aging. Remarkably, aging-dependent loss of LamC and the ectopic expression of LamDm0 are suppressed by over expression of Hey in ECs of aged flies (new Figure 7).

2) Expression of LamC or elimination of lamDm0 each one by itself, or in combination was not sufficient to rescue the phenotype of loss of Hey. Since Hey is situated above the two lamins and act also directly on Pdm1, this inability is not surprising.The transcriptional activation by Hey of EC genes is mediated by itself and Pdm1, and therefore is not compensated by expression of LamC or loss of LamDm0 that act only on the repressive armof Hey activity.

3) Moreover expression of LamC was not sufficient to inhibit the ectopic expression of Δ. we relate and describe this in the discussion: “Over-expression of LamC in Hey-targeted ECs was not, however, sufficient to repress ectopic Δ expression (data not shown), suggesting that the repressive activity of LamC may require the presence of Hey or a Heyregulated protein(s) on these gene regions.” Thus, Hey activity is more broad than just the regulation of nuclear lamins, it is likely required for lamin-dependent repression and its activity is essential for both repression and gene activation

Reviewer #2:

[…] Overall, I feel that the data presented--although substantial in volume--unfortunately fall far short of providing convincing support for these conclusions. In my view, the revisions necessary to substantiate the authors' claims would be quite a major undertaking.My concerns involve what I see as significant issues regarding data interpretation and study design:1) Hey acts broadly in multiple cell types, but the study does not offer insight into how Hey achieves cell type-specific regulation. Specifically, Hey is expressed in all midgut cells and has pleiotropic roles in stem cells, enteroblasts, and enterocytes. Without knowing the mechanisms that underlie cell type specificity, I am not sure whether Hey is actually a core transcription factor or merely an accessory co-factor.

We provide the following experimental evidence that Hey is a core transcription factor and not an accessory/redundant transcription factor:

1) Loss of Hey in ECs cannot be compensated by the expression Pdm1.

2) Increasing the amount of Hey by its Expression in stem cells represses the expression of ISC-specific genes like Δ and LamDm0 as well as likely promotes premature endoreplication. All features that are characteristics of differentiated ECs.

3) Hey regulated genes in ECs and in Progenitors are very different with only minimal overlap, suggesting for specific hey-regulated signature in the different cells. (Figure 3H)

4) As discussed for Rev.1 loss of LamDm0 or expression of LamC cannot suppress Hey phenotype. Taken together these data suggest that Hey has a key role in EC differentiation and is a pivotal transcription factor in maintaining EC identity. Thus, its activity is not “an accessory” cofactor, as suggested by the reviewer.

2) A major element of the authors' model is that the switch from LamDm0 to LamC during enterocyte differentiation produces widespread changes in gene expression by promoting different nuclear architectures. However, the study does not include any assessment of nuclear organization, lamina-chromatin association, or other lamin-influenced structural feature for any midgut cell type (either normal or genetically manipulated enterocytes and enteroblasts). As far as I can tell, the authors' conclusions that Hey and LamC "remodel the nuclear architecture" and "determine a nuclear organization that is unique to ECs", although provocative, are not supported by either direct or indirect evidence.

The reviewer’s criticism is correct. Therefore, in the current manuscript we have now extensively addressed this issue of large-scale organization of the EC nucleus upon Hey knockdown in ECs.

I) We show that loss of Hey in EC resulted in increased global chromatin accessibility in ECs (new Figure 4).

II) We show that loss of Hey affected global organization of EC nuclei (new Figure 4).

Specifically, we show:

I) “We therefore investigated whether loss of Hey in ECs results in a global change in chromatin conformation using an M. SssI methylation-based chromatin accessibility assay (Figure 4F, G; Rincon-Arano et al. 2012). In brief, the M. SssI enzyme efficiently methylates CpG dinucleotides in vitro depending on chromatin accessibility. This methylation is endogenously minimal in differentiated somatic Drosophila cells. The methylated dinucleotides are subsequently detected using 5mCmAb (Bell et al., 2010). Only minimal 5mC methylation was detected in control gut ECs (9%, n=206), but upon Hey knockdown, a significant increase in 5mC was detected in polyploid cells using immunofluorescence with the 5mC antibody likely found in ECNFD or PMD cells (22% n=406; p< 0.001) (Figure 4F, G and quantitated in Figure 4—figure supplement 3G)”.

II) “RNAi-mediated depletion of Hey resulted in impaired expression and distribution of heterochromatin protein 1c (HP1c), which was no longer localized to the chromocenter (Figure 4I, J). It also resulted in an increased distribution of Nop60B and Coilin, which mark the nucleolus and Cajal bodies, respectively (Figures 4K-N). We also observed impairment in the nuclear localization of LSM11 that is associated with the histone locus bodies (HLBs) (Figure 4O, P) and at the nuclear periphery, localization of Mtor, a nuclear envelope component, was distorted. Alongside changes in the expression of nuclear lamins (Figure 4Q-V). We observed reduced expression of Lamin C and increased expression of LamDm0. We also observed increased expression of the LamDm0 binding protein, Otefin, as well as its redistribution to the nuclear periphery (Figures 4W-X’, 4H). In contrast, the immunofluorescence signal of the co-repressor dSin3A, which functions in HESmediated repression, was identical in both Hey-RNAi- and control-targeted gut (Figures 4Y, Z, Figure 1—figure supplement 3K, L).”

3) The datasets that are used to draw conclusions about cell type-specific transcriptional signatures and enhancer activity are not themselves specific to the relevant cell types.a) To isolate enteroblasts for microarray analysis (Figure 4A), the authors sort cells for UAS-GFP under control of a GAL4 driver comprising the 3'UTR of hey. They state that this driver is expressed "predominantly" in enteroblasts. However, many Δ-expressing stem cells appear to express hey>GFP (Figure S4A), and many cells that express hey>GFP do not express the enteroblast-specific Notch reporter (Figure S4B). These features raise questions as to the specificity of the "enteroblast" datasets.

We agree with the reviewer that Hey-GAL4 drives expression also in ISCs (albeit to a much lesser amount that EBs). We therefore revised our statements and agree that this should be considered as a “progenitor driver”. We now relate to these cells a “progenitors” and not purified EBs. We have extensively modified the text and data to accurately present the experiments, as well as their interpretation.

b) Hey enhancer activity is inferred from DamID of cultured Kc167 cells, not of midgut cells. The authors state they were unable to perform DamID in midgut cells (although some precedents do exist: Korzelius et al., 2014, Loza-Coll 2014, Jin 2015, Chen et al., 2018). Given that Hey has different targets in different cell types even within the midgut, it seems tenuous to extrapolate enhancer binding in Kc167 cells to midgut enterocytes.

We agree with the reviewer, and due to the concerns raised and in the current manuscript we do not discuss a potential role of Hey on enhancers. We also limited the discussion regarding the results of the DamID experiment. However for few genes that show binding in DamID, that we experimentally validated in ECs, and are important for ECs identity we show binding to predicted enhancer regions. These include genes like Pdm1, Δ, LamDm0, and LamC. Experimentally, we also show that the M5-4 enhancer reporter line of the stem cell esg gene that is active only in stem cells is ectopically activated in ECNFD (Figure 2 and Figure 5).

4) It is not clear to me that hey-knockdown enterocytes truly "lose their differentiated identity", as stated by the authors and as would be expected if Hey is a core terminal factor.a) Morphologically, the knockdown enterocytes still resemble enterocytes and not any other cell type.

While morphologically they still resemble polyploid ECs, a long the manuscript we bring evidences both by transcriptional analysis as well and using G-TRACE-confocal analysis that ECNFD fail to express many ECs specific genes including the key transcription factor Pdm1, EC specific adhesion molecules (Figure 2), and many putative digestive enzymes (S6). Moreover, these targeted ECs exhibit abnormal cell cycle phasing likely re-entering aberrant endoreplication visualized in vivo using an RGB cell cycle tracer line. For simplicity we did not relate to changes in cell cycle in the manuscript (see an example at end of point by point).

b) Physiologically, not evidence that enterocytes downregulate digestive enzymes or nutrient transporters or otherwise attenuate their digestive function.

We now show in Table S6 that more that 50 putative digestive enzymes are not expressed upon loss of Hey in ECs. This data is based with an extensive detailed excellent analysis of gut transcriptomics. In addition GO analysis of Hey downregulated genes show classical EC physiological groups such as transporters lipases, metabolic enzymes etc (Figure 3B, Table S2). We also show aberrant lysosomal activity using lyzo-tracker, and loss of gut integrity.

c) The phenotypic effects of hey knockdown (e.g. elevated JAK-STAT, expression of Notch reporters, loss of H3K27Ac, lamin mis-expression) appear partial, which is inconsistent with the notion that Hey is centrally important. (These partial effects do not appear to correlate with the extent of Myo1A driver activity, as far as I can gauge by co-expression of GFP in the images that are shown.)

The phenotype of loss of Hey are actually extremely strong. The gut structure is strongly affected and hard to dissect even after 72-100 h. Therefore we performed experiments after 48h hours. Under these experimental conditions many cells but not all lose identity as shown nicely in G-TRACE experiments (Figure 2). However as in many cases with RNAi this is not a null. Indeed using MARCAM we show that Hey deficient clones are hardly generated and they do not mature into full differentiated ECs. Thus, Hey is critical for establishing and maintaining EC identity, we therefore believe that the impression of the reviewer in this case is not correct.

d) Many effects of hey knockdown resemble the well-characterized hallmarks of regeneration and ageing (e.g. elevated JAK-STAT, expression of Notch reporters, disorganization, leakiness). Indeed, I might interpret the hey knockdown phenotype as simply recapitulating the normal enterocyte response to damage and stress-not as wholesale reprogramming of enterocyte identity.

We agree with the above criticism and in the current manuscript focused on aging. Indeed, we noticed that Hey protein levels in ECs decline with age (Figures 7A, B). Moreover, the phenotypes resulting from acute conditional knockdown of Hey in young flies (2-4 days old) are highly similar to the phenotypes observed in wildtype guts derived from aged flies (3-4 weeks old). As seen in NEW Figures 7C-J, aging ECs exhibit a reduction in the MyoIA-GFP signal, a decline in Pdm1 and LamC expression, along with ectopic expression of Δ and LamDm0 and overall reduced gut integrity (Figures 7C, 7E, 7G, 7I, 7K). Indeed, continued expression of Hey in aged ECs using MyoIAts-Gal4 prevented the ectopic expression of Δ and LamDm0 (Figures 7D, 7J), restored the expression of Pdm1 (Figures 7E, 7F), of LamC (Figures 7G, 7H), and of overall EC morphology, compared with control. Finally, expression of Hey prevented loss of gut integrity as revealed by the “smurf” assay (Figures 7K, 7L).Thus, we bring novel physiological role for Hey in maintaining aged dependent loss of EC identity.

[Editors’ note: what now follows is the decision letter after the authors submitted for further consideration.]

Reviewer #2:

The manuscript of Flint-Brodsly et al. is an exciting manuscript. The authors have improved the flow of the manuscript and strengthened their conclusions, by performing multiple additional experiments, while correcting the phrasing of certain points throughout the manuscript at the same time.The manuscript describes the pleiotropic role of Hey TF in the Drosophila gut, focusing mostly on its role in the active maintenance of the differentiated state of the EC cells. The techniques the authors used range from genetic and cell biology experiments to multiple RNAseq and DamID experiments. The authors have addressed this reviewer’s remarks.Additional data files and statistical comments:A major problem that I have in evaluating this work is the general lack of quantification throughout the manuscript. The adult gut is a very dynamic tissue that responses rapidly to subtle environmental changes. Because of this, it is especially important that careful quantification be performed as it is very rare that any phenotype is black and white. Therefore, while there are some potentially interesting findings (changes in Lamins during differentiation; a role of Hey in EC LamC expression; potential regulatory roles for Lamins in controlling fate genes), it is impossible to judge from the data presented whether the anecdotal images are representative of all guts, all regions of the gut, males or females, etc. Ideally, all data should be rigorously quantified, with data points plotted - not just an average-- and compared with appropriate statistics.

As detailed below we performed all the quantifications and statistics as suggested. As stated in the text, we have performed all the experiments in females. In addition, in the revised version we added analysis of Hey loss also in males (new Figure 1—figure supplement 2A, B), and show the impact of Hey loss on the entire gut (new Figure 1—figure supplement 2C-D’). Rigorous quantification is now presented along the entire manuscript.

In particular, the effect on Pdm1 expression of Hey knockdown with Myo1AGal4 looks more subtle than some other phenotypes, which raises the issue of how direct or indirectly certain phenotypes are. Proper quantification of the data will make this more clear.

We quantified the data regarding Pdm1 at the protein level, and this is now shown in new Figure 1—figure supplement 1E. Please note that the reduction in Pdm1 expression was observed using different methodologies (global, and G-TRACE analyses of knockdown of Hey). Pdm1 was identified as a putative direct target of Hey by DamID (Hey binds to its predicted enhancer).

Another major issue that I have is that because the RNAi of Hey in ECs generates a proliferative response, many of the effects the authors detect could be secondary due to this. This should be addressed.

We fully agree. This is now better addressed including our G-TRACE analysis, as well as in the discussion. The potential cross-talk from ECs that lack Hey or LamC to progenitors or during aging is a broad topic that we are studying. I feel that the experimental molecular details; such as the identity of ligands secreted from ECs and that impinge and induce on progenitor regeneration is outside of the scope of this current manuscript (that is already dense), as these experiments will require at least six months (obtaining stocks and the subsequent genetics required).

Finally, the paper could be more streamlined to improve readability. There are some things that seem irrelevant and could be removed (Figure 2E', Figure 6S, T). Also, it seems to me that the characterization of the proliferative response in Figure 6 needs to come much earlier in the manuscript as it is important to interpret the rest.

We improved the overall streamlining the paper. As suggested, we removed old Figure 6S, 6T (this experiment was added at the request of a reviewer in the first cycle of *eLife* revision).

The "hyperplasia" is mentioned on p5 but not shown then or quantified.

As suggested we better present the data regarding hyperplasia including quantification shown in Figure 1J.

1) Quantification throughout, n= should be somewhere in legends or figures.

All the quantification and “n” numbers were added.

– Figure 1B-D. What% of cell types are marked by Hey?

Hey is expressed in all gut cells with high levels in EBs and ECs. We now added new Figure 1—figure supplement 1K-P that show the DAPI channel alongside endogenous Hey immuno-staining and cell specific markers demonstrating this and complementing the data presented in Figure 1.

– Figure 1E, E', G, G': How frequent is the large Dl+ cell phenotype. One example is shown- please quantify% of Dl+ cells that are Pdm1+.

We observed that 7-15%, of the polyploid cells to be positive for Dl and negative for GFP upon Hey knockdown this is now shown in Figure 1J’. Such cells are never observed in control. Unfortunately, double staining of Δ and Pdm1 cannot be performed as both antibodies are mouse monoclonal.

This phenotype might be linked to the proliferative state of the gut producing more newly made ECs. Their data do not show that the EC which downregulates Hey actually re-expresses Dl. Another hypothesis is that Hey knockdown in ECs triggers stress and activates ISC proliferation, in which conditions ISCs are known to be enlarged (for example upon yki overexpression, see Shaw et al., 2010, Figure 1H-J).

In the revised version we better clarified this point and included the relevant reference. as correctly stated by the reviewer, polyploid cells (PPCs) that do not express GFP or RFP are likely mis-differentiated progenitors expressing Δ. Importantly, that G-TRACE analysis presented in Figure 2E’ shows that it is the PPCs that are GFP (+) and RFP (-) (PPC **) that express Δ on their surface. Based on the G-TRACE technique and since the expression of GFP is the result of a recombination event driven by active MyoGal4 (an activation that takes place only in the differentiated ECs), it is highly suggestive that PPC GFP(+) RFP(-) were EC initially, are currently not maintaining EC identity (hence are RFP negative) but express the GFP history marker.

– Figure 1F, F', H, H4- what% of ECs lose Pdm1 staining up Hey RNAi expression compared to controls?

We observed that 82% of cells lose Pdm1expression upon targeting Hey. This is now presented in new Figure 1—figure supplement 1E (p<0.0001, n (h) 784, n(c) 271).

Can they co-stain with Hey to see if there is a direct relationship between Hey disappearance and Pdm1 disappearance?

We tried to perform the co-staining but technically this was not successful probably due to difficulties with both antibodies

– Figure 2 F, Figure 5I: the n should be put in the figure.

“n” entered into the figure.

– Figure 3 D-G: What% of Dl+ cells express RecQ4 in Myo1A>Hey RNAi and please compare to controls.

We found that 17% of PPCs co-express Dl, RecQ4. These cells are not observed in control guts. A quantification is now shown in new Figure 3G’.

– Figure 4C, C'; Figure 5A-B- Importantly a control is missing: the same region of guts expressing no RNAi. Due to the regional differences, I am not convinced one can compare across compartment boundaries like in the wing disc. Please add control and quantify% of cells altered for both.

Controls of regional targeting of Hey are now shown in Figure 4—figure supplement 2D, E and Figure 5—figure supplement 1A, B, and quantitative analysis in Figure 5—figure supplement 1C. In addition, we presented similar results using the MyoIA-Gal4ts system that acts globally in the midgut; demonstrating that loss of Hey resulted in reduction of H3AK27ac, LamC and increased expression of lamDm0 (see new Figure 4—figure supplement 3H-K, Figure 5—figure supplement 1F-I). As suggested Quantification of all the regional experiment was performed in comparison to targeted region of control and is now shown in Figure 5—figure supplement 1C.

– Figure 4D-G: These phenotypes could be due to differences in proliferation between HeyRNAi and wild-type. Newly produced ECs will inherit LacZ and may have more accessible chromatin. What does the LacZ line look like in wild-type MARCM clones that would similarly be proliferative tissue? In any event, these phenotypes need quantification.

We agree with the reviewer, addressed this in the text, and show quantification in Figure 4—figure supplement 3E-G.

– Figure 4I-CC; Figure 5G-Q: Please show the% of cells show these phenotypes, compared to controls, with stats.

As suggested we added a detailed quantitative analysis with statistics. This is now shown in a table in new Figure 4—figure supplement 3L

– Figure 6: some quantification must be done here. The effect on MESH looks much stronger than the effect of Hey knockdown, in terms of% of ECs losing Hey. Why is this? Quantification of PH3 with >15 guts.

Quantification of phosphoH3 is now shown Figure 6O. The impact on MESH is strong and maybe due to collapse of the entire structures between cells that impact MESH stability.

– Figure 6UV: They state in the legend that 22% of flies have a smurf phenotype upon Hey RNAi. It would be helpful to write this on the figure so it is clear.

% is now written within the figures.

2) An important part of the manuscript is the finding that Lamin types switch from the progenitors to the EC and that Hey is required for maintenance of expression of the LamC in the EC. This finding seems to directly contradict recent data of Petrovsky and Großhans, 2018 that observe that LamC is uniformly expressed in the epithelial cells of the Drosophila intestine. What is the difference in experimental set up or interpretation? The authors should minimally discuss this contradictory finding that goes against their model.

This paper was published after our manuscript was sent to review and therefore was not cited. In this work the authors identified that the expression of LamC in all cells based on the correlation with progenitor cells expressing the Esg-gal4 driver. This driver marks both EB and ISCs, but does not discriminate between them. This is in agreement with our observations that clearly show high levels of LamC in progenitors; low level in ISC and higher in EBs see Figure 4CC.

In the current version we relate and cite their work: “As our manuscript was under review Petrovsky and Großhans also characterized the expression of nuclear lamins in the midgut (Petrovsky and Großhans, 2019). Similar to our observations they reported that LamDm0 and Otefin are enriched in progenitor cells. They also reported the expression of LamC in progenitors cells. Our work using ISC and EB specific markers further established that in progenitors high levels of LamC are present in EBs and only minimal amount is expressed in ISCs.”

3) Important interpretations were made using GO term analysis. I am concerned that the results obtained are biased toward genes expressed in the ECs. My understanding is that when performing GO term analysis, the background gene set should be the expressed genes in the tissue or cell type. If this is not used, a bias will be obtained towards these genes since, the genes altered are those expressed in the tissue. The authors should use the background gene set from the gut epithelia plus surrounding muscle, which can be obtained at Flygut-seq for the analysis in Figure 3 and S8 to see if the categories of genes altered in HeyRNAi or LamDm0 overexpression are still statically enriched over the gene set of intestine genes (and not the whole genome).

As suggested we compared our gene expression data to the Flygut dataset to a “top hits per region” available at: https://flygut.epfl.ch/expressions. We find that the categories of genes altered in Hey RNAi or LamDm0 overexpression are still statically enriched over the gene set of intestine genes. We added this new data along the text and present it in Figure 3—figure supplement 4Q and Figure 5—figure supplement 3E.

4) I feel that the authors overinterpret the data presented in Figure 2A-I'. It is also written in a very confusing manner- please start by outlining the experiment and the expected results, then give the results without an interpretation of what the different colors of cells are. Then speculate on the potential meaning. They should not refer to the GFP+RFP- cells as "no longer fully differentiated". Similarly, do not refer to the RFP+GFP- cells as "early ECs" or the GFP-RFP- cells as "potentially mis-differentiated". These interpretations should be kept separate from the data and then substantiated with additional markers.

We agree with this correct suggestion and critic. We extensively revised this section in the text and relevant figures and show analysis with markers. we now call these cells PPCs (polyploid cells). We now refer to the GFP+RFP- cells as PPC**, and GFP- RFP- as PPC* and only at the end of the section we suggest interpretation based on the G-TRACE system.

A timecourse would be helpful as these populations should evolve over time.

As suggested we show G-TRACE analysis within 24h and 48h time points. Quantification of this is presented in Figure 2—figure supplement 2A’-A”.

Also, given the difference in proliferation of control and knockdown, could the results be explained by altered kinetics in a regenerating gut? They could see if infection or damage conditions show a similar effect, which would argue against their model.

Our manuscript focused on how ECs maintain identity, but rather in young adults or during normal aging. We strongly feel that how cell identity is maintained upon infection or DNA damage – while highly interesting – is outside of the scope of the present study.

– Similarly, I believe they have misinterpreted the phenotype of Hey RNAi with respect to Notch signaling. Discussion, "The ectopic activity of a Notch-reporter observed in EC-NFD upon Hey targeting in ECs suggests that Hey limits Notch activity in ECs." This is likely only due to a secondary effect on causing stress in the gut, driving ISC proliferation, and therefore making more, new EB cells that express the Su(H)-GBE reporter. It is unclear in the gut whether Hey has any impact on Notch signaling, is regulated by Notch, or is simply parallel.

The reviewer makes a good point. We have changed the text to better reflect this point:

“The observed Notch-reporter activity in PPCs upon Hey targeting in ECs, may reflect stress-increased ISC proliferation and Notch activation in enlarged progenitors. Alternatively, this ectopic activity may suggest that Hey limits Notch activity in ECs.”.

5) Did the authors show data of LaminC RNA decrease in S2 cells?

No, we have not analyzed mRNA levels of nuclear lamins in S2 cells (LamDm0 or LamC).

6) "These phenotypes were suppressed by over-expression of UAS-Hey, but not by a control transgene (Figures S1C, C4). Moreover, RNAi knockdown of Notch, Suppressor of Hairless, other HES TFs, and HES-related cofactors did not phenocopy Hey loss in ECs (Figures S2A-J)."– what phenotypes are being assessed. Please show some quantification.

We explained better the phenotypes in the text and showed quantifications (Figure 1J, J’).

7) It should be clearly stated in the text that microarrays are used to assess transcriptomes. Quantification of the numbers of hey-Gal4 expressing cells should be done in the different backgrounds (controls vs heyRNAi). PCA plots should be shown.

This is now stated in the Results section and not only in Materials and method section. PCA plots are presented in new Figure 3—figure supplement 4D, E and Figure 5—figure supplement 4.

8) P7 "Overall, the transcriptional signature of guts in which Hey was targeted in ECs largely resembled the signature of control purified progenitors"– "Hey knockdown was induced" would be more clear

As suggested we replaced this statement

Also, it is not at all clear to me what is being compared here: is it the genes that change in Myo1A>GFP vs Myo1A>hey RNAi compared to the genes that are expressed in "heyGal4+ cells"? What are the heyGal4+ cells compared to? Whole guts? These are primarily EBs. The interpretation of these data are not clear to me? The microarray of whole gut while knocking down in ECs will give ISC, EB and EE genes altered in a non-autonomous fashion as well as EC genes altered.

We now better explain the gene expression experiment in the text. We also describe the potential alteration in gene expression due to non-autonomous effects.

9) As mentioned above, quantification of the effects seen in Figure 4F-G; What% of guts show the effect? Figure 4 I-CC, what% of ECs show these effects?

As mentioned above we quantified all these phenotypes see Figure 4—figure supplement 3L.

There is undoubtedly variability in these data- what is the level? Could these effects be due to different ages of EC cells? More new EC cells are likely made in Hey knockdown conditions- is it only new EC cells that show this? They could for example make MARCM clones and look at 3 day old ECS in wild-type to see if the chromatin marks are established or if they come during maturation of the EC.

We agree with the reviewer’s point. However, it is still the case that Hey regulates chromatin accessibility during EC maturation and in its absence, chromatin is more accessible and adopts a more “open” structure. These effects are now quantified in Figure 4—figure supplement 3E-G.

10) Figure 5G-I- they should also quantify what happens to Dl+ cells- here they lose both Dl+ cells and LaminDm0+ esg+ cells.

This quantification is now presented in revised Figure 5I where it is shown that only 10% of progenitors (Esg>GFP+) that express Hey co-express Dl and LamDm0 compare to 100% in control

11) The data in 5J are not convincing on their own. This does not look like Dl staining, which is usually vesicular. Please show some additional images and quantification. Similarly, 5M is not convincing: in this context, clearly there are more ISC divisions, leading to more esg+ cells, which are likely not all ISC/EBS, but newly produced ECs, which therefore express Pdm1. How do the authors identify progenitors in Figure 5M?

We replaced these figures with better representative images and show quantification in Figure 5T-W.

14) PCA plots of the RNAseq data should be shown.

PCA data is shown in Figure 3—figure supplement 4D, E and Figure 5—figure supplement 4.

They should state to what they are comparing the Myo1A>LamD0 data (UAS-GFP?).

Indeed, this is the case and it is now clearly stated in the text.

15) I disagree with the following statement in the Introduction and Discussion, that I do not feel is supported by their references:"Indeed, multiple lines of evidence have established that failure to maintain a differentiated identity is a hallmark of aging…" (Schwitalla et al., 2013; Deneris and Hobert, 2014; Ocampo et al., 2016)To me, none of these references really strongly support the idea that differentiated cell fates decline during normal aging, which is what stated as fact. I would not say that this is a well-established notion. However, one paper that does support this is from the fly gut field and is not cited (Li, Qi and Jasper, 2016).

As suggested we revised the text: “Indeed, failure tomaintain a differentiated identity is associated altered physiological properties of post-mitotic cells and tissues resulting in disease such as diabetes, neurodegeneration, and cancer (Deneris and Hobert, 2014; Ocampo et al., 2016). We also added the relevant reference (Li, Qi and Jasper, 2016)”.

– In the Discussion section on Cell identity, aging, and pathological plasticity. "During Drosophila and vertebrate aging, or under various stress condition, differentiated cells in the gut, pancreas, and trachael epithelium behave similarly, reflecting a general principle." While, I agree published data suggest plasticity in cancer or pathological cell loss in these tissues, a function of loss of identity during normal aging is less obvious to me and should be better supported by published literature. Mutant contexts such as Progeria syndromes are not equivalent to wild-type aging.

We revised the text:

“One well-studied case is that of human LamC ortholog, LamA, which is mutated in Progeria, a disease associated with premature aging and loss of physiological properties of differentiated cells and tissues (Hegele, 2003). However, further studies are needed in order to link loss of identity and premature aging in progeria models and patients, to the consequences of physiological aging. Therefore, the identification of conserved mechanisms underlying the regulation of identity networks in model organisms such as *Drosophila* has broad implications for addressing pathological conditions in humans, and specifically age-related diseases.”

[Editors' note: further revisions were requested prior to acceptance, as described below.]

Reviewer #3:

“The proposed experiment at the end of this reviewers critique seems easy to do and should be informative”. “What I was suggesting that the authors do was not a 6 months experiment, but a simple treatment of flies with some damaging agent that induces a similar proliferative response to that seen upon Hey RNAi in ECs (one of: Ecc15, Pe, paraquat), and assess some of their nuclear phenotypes (LamC, LamDm0). If they do not see an impact on these marks, this would definitively rule out that the effect of Hey knockdown on ECs is merely indirect through rapid production of new ECs…I do realize that this is extra work, but honestly, if this were my own paper, I would at least want to know this result and comment on whether or not this is likely indirect via proliferation stimulation.

As suggested we performed the experiment that is now shown in new Figure 5—figure supplement 2I-P. We have evaluated the changes in expression of LamC, lamDm0, and Mtor upon exposure to paraquat, that induces a rapid proliferative response.

We have performed this experiment using MyoIA >GFP marking ECs, and similar to the experiment where we knockdown Hey using Hey RNAi. Paraquat treatment resulted in progenitors proliferation and formation of polyploid cells. However unlike in the case of targeting Hey, or during aging, the ECs in the gut retained the expression of MyoIA-GFP as well as LamC. They did not ectopically express LamDm0. Nor did they show changes in the expression of Mtor. Thus, this experiment strongly suggest that the effect observed upon loss of Hey is directly affecting ECs, and not merely reflecting only proliferative stimulation of progenitors.

We now added in the text:

“Importantly, the changes observed in LamC, LamDm0, expression and nuclear organization (such as localization of Mtor), were not observed upon paraquat exposure, that induces rapid progenitor proliferation (Figure 5—figure supplement 2I-P; Chatterjee et al., 2009). Thus, these changes are directly related to loss of Hey supervision in ECs”.